# Reciprocal control of excitatory synapse numbers by Wnt and Wnt inhibitor PRR7 secreted on exosomes

Sang H. Lee [1,2], Seung Min Shin[1], Peng Zhong[1], Hyun-Taek Kim[3], Dong-Il Kim[1], June Myoung Kim[4], Won Do Heo[5], Dae-Won Kim[4], Chang-Yeol Yeo[6], Cheol-Hee Kim [3] & Qing-song Liu[1,2]

Secreted Wnts play crucial roles in synaptogenesis and synapse maintenance, but endogenous factors promoting synapse elimination in central neurons remain unknown. Here we show that proline-rich 7 (PRR7) induces specific removal of excitatory synapses and acts as a Wnt inhibitor. Remarkably, transmembrane protein PRR7 is activity-dependently released by neurons via exosomes. Exosomal PRR7 is uptaken by neurons through membrane fusion and eliminates excitatory synapses in neighboring neurons. Conversely, PRR7 knockdown in sparse neurons greatly increases excitatory synapse numbers in all surrounding neurons. These non-cell autonomous effects of PRR7 are effectively negated by augmentation or blockade of Wnt signaling. PRR7 exerts its effect by blocking the exosomal secretion of Wnts, activation of GSK3β, and promoting proteasomal degradation of PSD proteins. These data uncover a proximity-dependent, reciprocal mechanism for the regulation of excitatory synapse numbers in local neurons and demonstrate the significance of exosomes in inter-neuronal signaling in the vertebrate brain.

[1] Department of Pharmacology and Toxicology, Medical College of Wisconsin, 8701 Watertown Plank Road, Milwaukee, WI 53226, USA. [2] Neuroscience Research Center, Medical College of Wisconsin, 8701 Watertown Plank Road, Milwaukee, WI 53226, USA. [3] Department of Biology, Chungnam National University, Daejeon 34134, Republic of Korea. [4] Department of Biochemistry, Yonsei University, Seoul 03722, Republic of Korea. [5] Department of Biological Sciences, Korea Advanced Institute of Science and Technology, Daejeon 34141, Republic of Korea. [6] Department of Life Science and Research Center for Cellular Homeostasis, Ewha Woman's University, Seoul 03760, Republic of Korea. Correspondence and requests for materials should be addressed to S.H.L. (email: shlee@mcw.edu)

Activity-dependent regulation of synapses is crucial for neural circuit development, maintenance of synaptic balance, and synaptic plasticity[1,2]. Aberrant regulation of synapse numbers is associated with numerous pathological conditions including autism-spectrum disorders, schizophrenia, and neurodegenerative disorders[3,4]. Notably, overall synapse numbers in mature neurons remain relatively stable despite on-going activity[5], suggesting the existence of unidentified maintenance mechanisms that counteract the pressures of activity-driven synaptogenesis or elimination.

Wnts are powerful secreted factors promoting synaptogenesis during development, but are also necessary for synapse maintenance in adult nervous systems[6,7]. Several species of Wnts are expressed by principal neurons in the hippocampus and activity-dependently released by neurons[8–11]. Exogenous application of Wnt5a or Wnt7a promoted the formation and strengthening of glutamatergic synapses in mature neurons[10–13]. However, molecular mechanisms by which the synaptogenic activity of Wnts is regulated are poorly understood. During development, Wnt activity is countered by secreted inhibitory factors such as secreted frizzled-related proteins (Sfrps) and Dickkofp-1(Dkk1) that directly bind and sequester Wnts and Wnt co-receptor low-density lipoprotein receptor-related proteins (LRP 5/6)[14]. However, the expression of these Wnt inhibitors in mature hippocampus is very low under normal conditions, except for Sfrp3 and Dkk3 in the dentate gyrus granular cells[15,16], and more importantly, their involvement in the regulation of synapses of mature neurons is unknown.

Exosomes are one type of secreted extracellular vesicle and originate from the release of intraluminal vesicles of multivesicular bodies (MVBs) upon their fusion to the plasma membrane (PM)[17]. While originally thought as a garbage-disposal mechanism for cells, recent studies indicate that exosomes carry a variety of signaling molecules including proteins, mRNAs, microRNAs (miRNAs), and lipids. Furthermore, it was shown that secreted exosomes are absorbed by recipient cells either by fusion with PM or via internalization[17,18]. Therefore, exosomes have a potential to deliver cargo molecules to target cells. Interestingly, it was shown that cultured cortical and hippocampal neurons also release exosomes at the dendrites and soma[19,20]. Furthermore, active Wnts are also secreted on exosomes at the neuromuscular junctions of *Drosophilia*[18,21]. However, the function of exosomes in the vertebrate brain, especially in interneuronal signaling, has remained uncertain[22].

Proline-rich 7 (PRR7) is a protein first identified from the proteomic analyses of postsynaptic densities (PSDs) and is highly expressed in the cortex and hippocampus of adult brains[23]. PRR7 has a single transmembrane (TM) domain, followed by proline-rich amino acid sequence, and C-terminal PDZ-binding motif that can interact with PSD-95. However, the role of PRR7 in synapse regulation remains unclear[24]. In this paper, we describe a novel function of exosome-secreted PRR7 in the control of excitatory synapse numbers in central neurons.

## Results

### Activity-dependent secretion of PRR7 on exosomes by neurons.
When overexpressed in dissociated rat hippocampal cultured neurons, hemagglutinin (HA)-tagged PRR7 showed, in addition to somatodendritic distribution, peculiar immunostaining patterns of fine vesicular objects (speckles) outside the β-galactosidase (β-Gal)-filled transfected neurons (Txf) (Fig. 1a). Endogenous PRR7 also showed similar immunostaining patterns (Fig. 1a). Some of the PRR7 immunofluorescent speckles were frequently found in areas non-overlapping with somatodendritic (MAP2) and axonal (Tau) markers and did not colocalize with

PSD-95 (Supplementary Fig. 1a, b), suggesting the secretion of PRR7 to the extracellular space. Since PRR7 has a putative TM domain[23] and is primarily associated with membrane fractions of the forebrain (Fig. 1g), we thought of the possibility that PRR7 is secreted on extracellular vesicles[17]. Indeed, PRR7 is highly enriched in the exosome preparations ($100,000 \times g$ pellet, termed P100; Fig. 1b) purified from the culture supernatant (CS) of hippocampal neurons[25]. Further analyses of the P100 by sucrose gradient centrifugation (cfg) showed the co-floatation of PRR7 with exosome markers including Flotillin-1, Alix, and Lamp2[17,20] in fractions with equilibrium densities of 1.12–1.15 g ml$^{-1}$ (Fig. 1c). Intriguingly, Wnt5a and Wnt7a showed identical fractionation patterns to PRR7 (Fig. 1c), suggesting that Wnts are also secreted on exosomes by central neurons. Ultra-structural analyses of the PRR7-enriched fraction (#8) by electron microscopy (EM) revealed cup-shaped vesicles with average diameters of $98.82 \pm 3$ nm ($n = 100$; Fig. 1d). These biophysical and morphological features of PRR7-containing vesicle fractions conform to the typical characteristics of exosomes[17]. Finally, immunogold labeling experiments of exosomes prepared from the CS of HA-PRR7-transfected neurons using anti-HA antibodies confirmed the presence of PRR7 in exosomes (Fig. 1e and Supplementary Fig. 1c).

We next determined whether the exosomal secretion of PRR7 is a constitutive or regulated process. PRR7 amounts in exosomes were greatly reduced by *N*-methyl-D-aspartate (NMDA) receptor antagonist (2-amino-5-phosphonopentanoic acid; APV) but not by α-amino-3-hydroxy-5-methyl-4-isoxazolepropionic acid (AMPA) receptor antagonist (6-cyano-7-nitroquinoxaline-2,3-dione; CNQX) or GABA$_A$ receptor antagonist (bicuculline; Bic) treatment of the neuron culture (Fig. 1f), indicating that exosomal secretion of PRR7 is dependent on NMDA receptor activity. Moreover, activity blockade by tetrodotoxin (TTX; Na$^+$ channel blocker) greatly decreased PRR7 level in exosomes (Supplementary Fig. 1d). Importantly, the protein levels of Flotillin-1 in exosomes were not decreased by APV or TTX. Therefore, PRR7 is a cargo protein secreted on exosomes and is unlikely to be directly involved in the biogenesis or secretion of exosomes.

PRR7 expression in the cortex and hippocampus of mouse brain is gradually increased during development and reaches its highest levels at adult stage (Fig. 1h), suggesting that PRR7 functions not only in developing but also in mature brains. In contrast, PRR7 expression in the cerebellum gradually decreases during early development and is undetectable in adult cerebellum. In adult brain, PRR7 is expressed at its highest levels in the dentate gyrus of the hippocampus, and also abundant in the cortex and striatum (Fig.1i). Importantly, PRR7 immunoreactivity is observed only in Neuronal nuclei (NeuN)-positive cells and absent from glial fibrillary acidic protein (GFAP)-positive cells, suggesting its neuron-specific expression (Fig. 1j). These data suggest that PRR7 is a neuronal protein secreted on exosomes in an activity-dependent manner.

### PRR7 eliminates excitatory synapses in local neurons.
To investigate the potential role of PRR7 in the regulation of synapses, we first examined how its overexpression affects excitatory synapses in cultured rat hippocampal neurons by immunocytochemistry (IC). Overexpression of PRR7 in hippocampal neurons for 18 h drastically decreased the average puncta densities and intensities of excitatory synapse markers including PSD-95, vesicular glutamate transporter 1 (vGLUT1), membrane-associated guanylate kinases (MAGUKs), and SAP90/PSD-95-associated proteins (SAPAPs) in Txf (Fig. 2a–d and Supplementary Fig. 2a–f), suggesting that PRR7 eliminates excitatory synapses. Since PRR7 is secreted on exosomes, we also examined

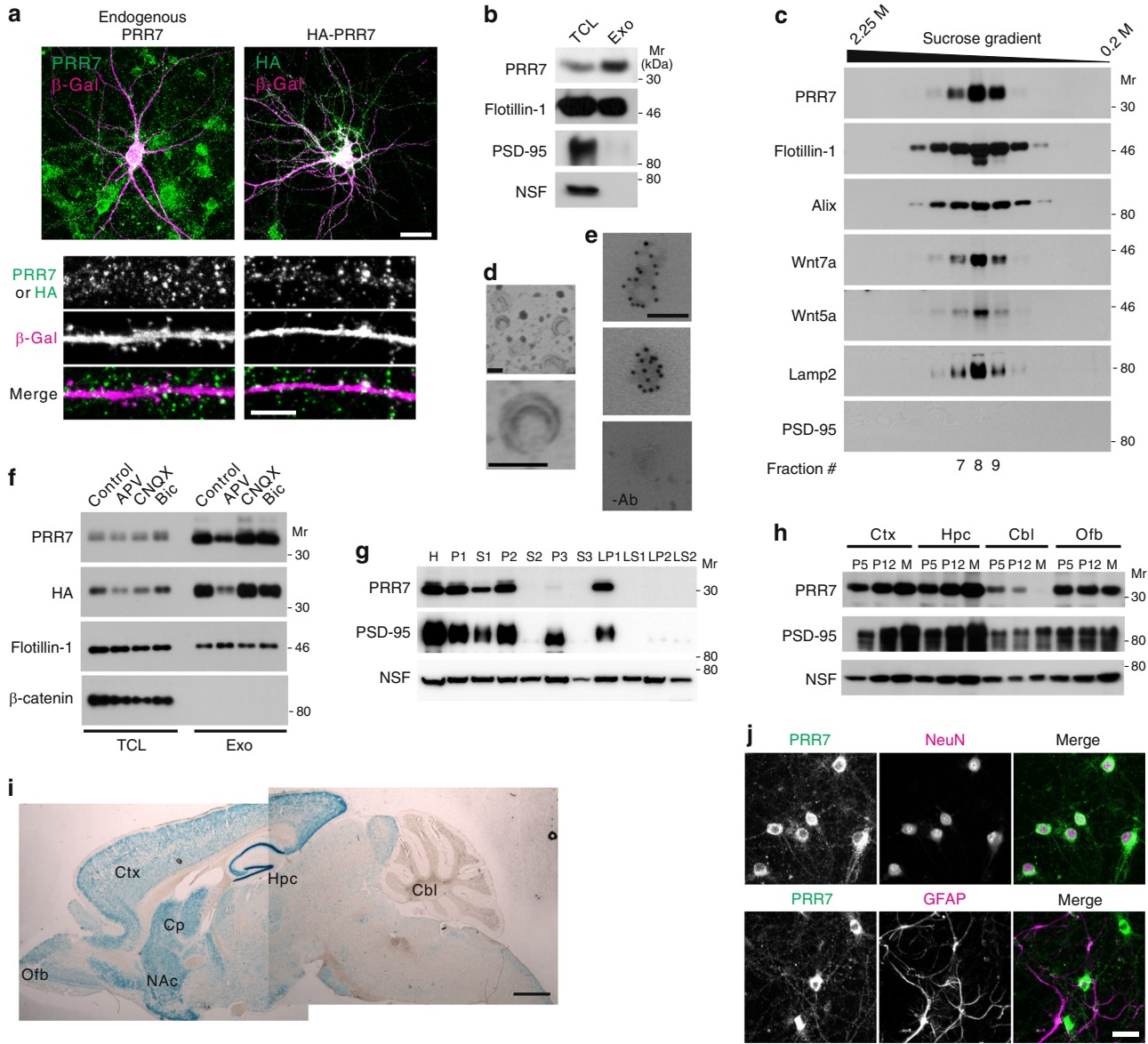

**Fig. 1** Exosomal localization and expression patterns of PRR7. **a** Immunofluorescent images of endogenous PRR7 (left) and transfected HA-PRR7 (right) in dissociated hippocampal neuron culture. β-Gal was used as a neuronal-fill marker. Scale bars, 20 (top images) and 5 μm (bottom images). **b** Enrichment of PRR7 in exosome preparation purified from the supernatant of hippocampal neuron culture. Exo exosomes, TCL total cell lysate. **c** Sucrose density gradient centrifugation analyses of PRR7-containing exosomes. **d** Electron micrographs of the PRR7-enriched fraction (#8) showing the typical cup-shaped vesicular structure of exosomes. **e** Immuno-EM images showing HA-positive immunogold particles on exosomes (top two panels; showing two exosomes and single exosome, respectively). A negative control image obtained in the absence of antibody (−Ab) is shown at the bottom. Scale bars (**d**, **e**), 100 nm. **f** Activity-dependent secretion of PRR7 on exosomes. **g** Distribution of PRR7 in various subcellular fractions of mature rat forebrain tissues. H total homogenate, P1 cell debri and nuclear pellet, S1 postnuclear supernatant, P2 synaptosomal fraction, S2 supernatant of P2, P3 microsomal pellet, S3 soluble fraction, LP1 synaptosomal membrane-enriched pellet, LP2 synaptic vesicle-enriched fraction, NSF *n*-ethylmaleimide-sensitive factor. **h** Developmental changes of PRR7 protein expression in various regions of the rat brain. Ctx cortex, Hpc hippocampus, Cbl cerebellum, Ofb olfactory bulb, P5 postnatal day 5, M 3-month old mature rats. **i** Expression pattern of *PRR7* as determined by β-Gal staining of brain sections from adult *PRR7* KO mice with β-Gal reporter knock-in. Cp caudate putamen, NAc nucleus accumbens. Scale bar, 1 mm. **j** Immunofluorescent images showing neuron-specific expression of PRR7 in rat hippocampal neuron culture. Scale bar, 20 μm. Mr relative molecular weight

the effect of PRR7 on excitatory synapses in neighboring non-transfected neurons (Nxf). Strikingly, PRR7 overexpression also reduced the number of excitatory synapses in Nxf (Fig. 2a–d and Supplementary Fig. 2a–f), suggesting that PRR7 has non-cell autonomous effects. In contrast, PRR7 with C-terminally fused green fluorescent protein (EGFP) (PRR7-GFP) showed reduced synapse-eliminating activity in Txf compared to HA-PRR7 and is virtually devoid of the non-cell autonomous effect (Fig. 2a–d and Supplementary Fig. 2b–c). On the other hand, PRR7 knockdown

by short hairpin RNA (shRNA)-mediated RNA interference (RNAi; for specificity see Supplementary Fig. 2g–k) produced the opposite effects on excitatory synapses (Fig. 2e–h and Supplementary Fig. 2l). In addition, co-overexpression of shRNA-resistant PRR7 (Supplementary Fig. 2g) completely abolished the RNAi effect on PSD-95 puncta (Fig. 2h), demonstrating PRR7-specific effect of the RNAi. Consistent with the loss of excitatory synaptic markers, PRR7 overexpression also greatly reduced the number of dendritic spines in Txf (Fig. 2i, j).

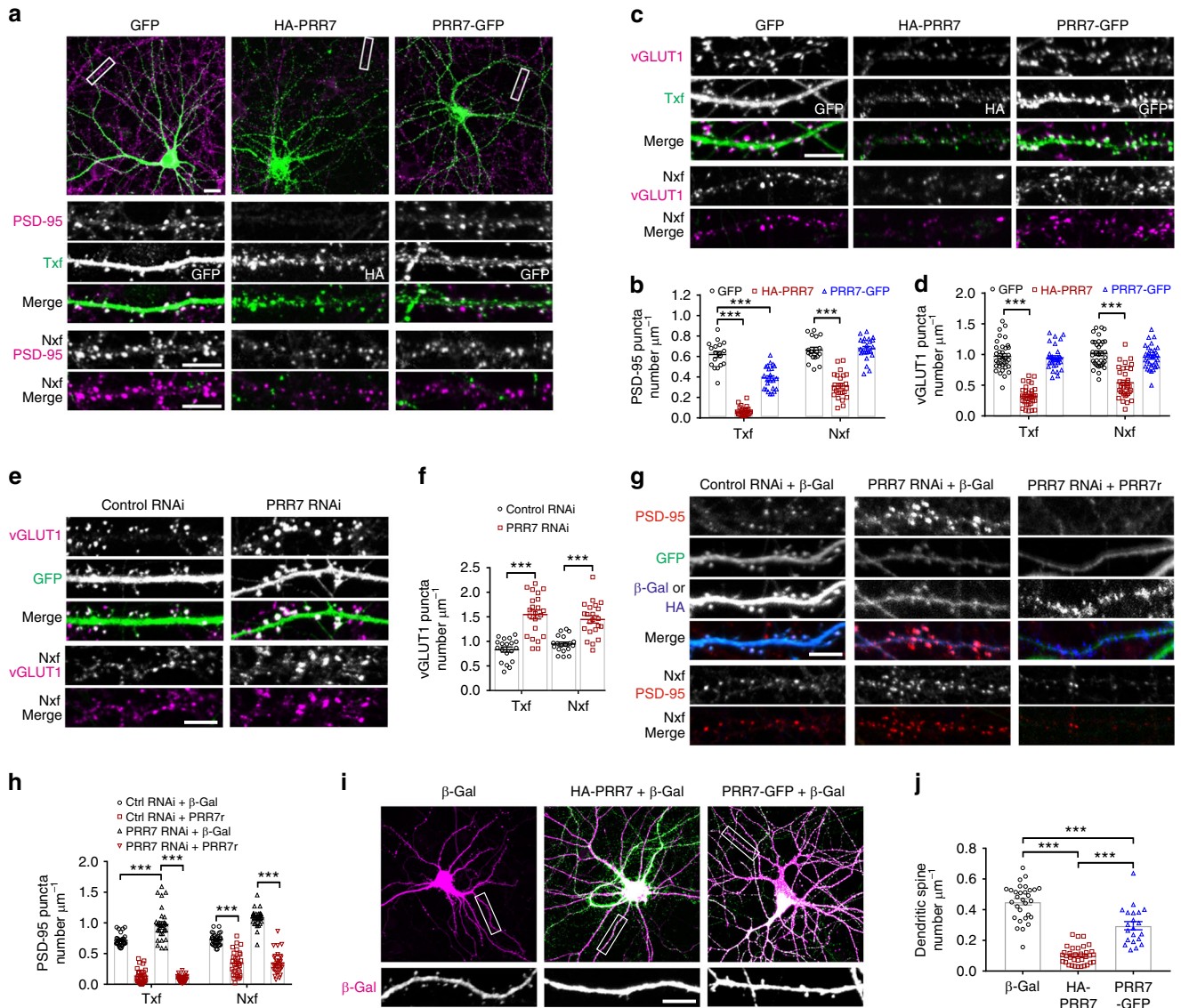

**Fig. 2** PRR7 controls excitatory synapse numbers. **a** Representative immunofluorescent images of PSD-95 in hippocampal neurons transfected with GFP, HA-PRR7, or PRR7-GFP. White boxes indicate areas shown in enlarged Nxf images. **b** Quantitation of average PSD-95 puncta densities. $n = 18$–24 neurons per condition. **c** Representative immunofluorescent images of vGLUT1 in hippocampal neurons transfected with GFP, HA-PRR7, or PRR7-GFP. **d** Quantitation of average vGLUT1 puncta densities. $n = 32$–35. **e** Representative immunofluorescent images of PSD-95 and vGLUT1 in hippocampal neurons transfected with PRR7 RNAi. **f** Quantitation of PRR7 knockdown effect on PSD-95 and vGLUT1 puncta densities. $n = 18$–22. **g**, **h** Loss of PRR7 RNAi effect by shRNA-resistant PRR7 (HA-PRR7r). Representative images (**g**) and quantification of the data (**h**). Ctrl control. $n = 30$. **i**, **j** The effect of PRR7 or PRR7-GFP overexpression on dendritic spine densities in transfected neurons. Representative images (**i**) and quantification of the data (**j**). $n = 18$. Scale bars, 5 µm. Data are mean ± SEM. Two-way (**b**, **d**, **f**, **h**) or one-way ANOVA (**j**), post hoc Tukey's test: $F_{2,124} = 207.6$ (**b**), $F_{2,194} = 148.1$ (**d**), $F_{1,78} = 76.06$ (**f**), $F_{3,232} = 380.7$ (**h**), and $F_{2,82} = 100.4$ (**j**). ***$P < 0.0001$

We next determined the effect of PRR7 on excitatory synaptic transmission by measuring miniature excitatory postsynaptic currents (mEPSCs) in hippocampal neurons. Corroborating the IC data, PRR7 overexpression or RNAi greatly affected excitatory synaptic strength and synapse numbers of both Txf and Nxf neurons, as shown by respective changes in the amplitude and frequency of mEPSCs (Fig. 3a–c). Interestingly, PRR7-GFP reduced both mEPSC amplitude and frequency in Txf, but had no effect on both mEPSC amplitude and frequency in Nxf (Fig. 3a–c). In contrast, a PRR7 mutant with the deletion of the last 4 amino acids (dC4) showed the weakest ability to reduce mEPSC amplitude in Txf and had no effect in mEPSC frequency in both Txf and Nxf (Fig. 3a–c), suggesting that these two PRR7 mutants have different functional deficits. To address the potential artifact of dissociated neuron culture system, we also tested the synapse-eliminating effect of PRR7 in organotypic hippocampal slice culture that maintains in vivo synaptic organizations. Remarkably, simultaneous dual recordings from both Txf and directly adjacent Nxf CA1 cells showed that PRR7 overexpression or knockdown greatly affected the evoked EPSCs not only in Txf but also in Nxf (Fig. 3d, e) in a similar manner to mEPSCs. Therefore, the non-cell autonomous effect of PRR7 is likely a physiologically relevant phenomenon.

**PRR7 does not affect inhibitory synapses of Nxf neurons.** PRR7 overexpression moderately reduced the number of inhibitory synapses in Txf, as determined by immunofluorescent staining of

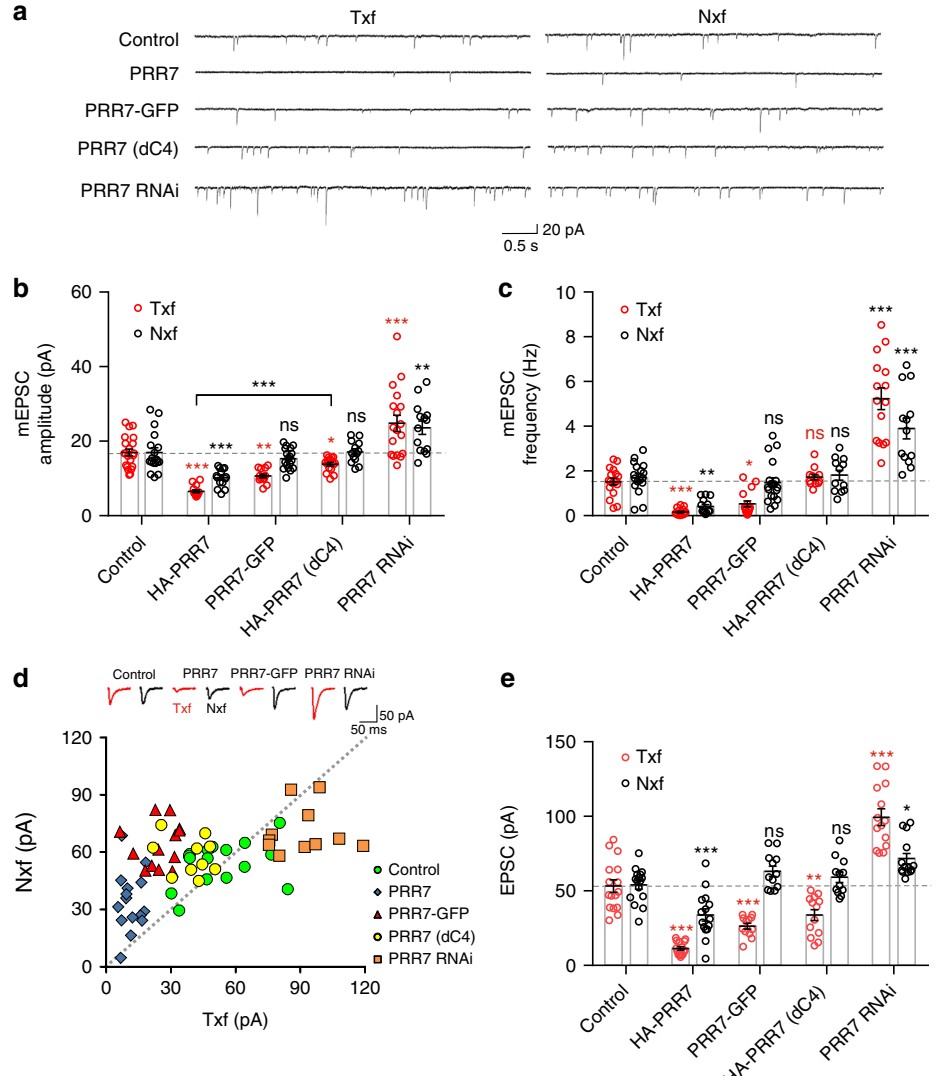

**Fig. 3** PRR7 regulates excitatory synaptic transmission of local hippocampal neurons. **a–c** Effect of various PRR7 constructs on AMPAR-mediated mEPSCs in dissociated hippocampal neuron culture. Representative traces (**a**) and quantification of mEPSC amplitude (**b**) and frequency (**c**). $n = 14$–21. **d, e** Effect of various PRR7 constructs on evoked EPSCs measured by simultaneous whole-cell recordings from Txf and Nxf neurons of CA1 pyramidal cells in organotypic hippocampal slice culture. $n = 12$–16. Data are represented as scattered plots showing individual data points and mean ± SEM. Two-way ANOVA, post hoc Tukey's test: $F_{4,150} = 46.2$ (**b**), $F_{4,143} = 85.45$ (**c**), and $F_{4,129} = 81.24$ (**e**). ***$P < 0.0001$, **$P < 0.001$, *$P < 0.01$, and ns not significant

inhibitory synapse markers, presynaptic vesicular GABA transporter (vGAT) (Fig. 4a–f) and postsynaptic GABA$_A$ receptor γ2 subunit (GABA$_A$Rγ2) (Fig. 4j–l). However, PRR7 overexpression effects on GABAergic synapses were not as drastic as those on excitatory synapses and completely absent in Nxf. Furthermore, PRR7-GFP overexpression and, more importantly, PRR7 knockdown had no effect on vGAT and GABA$_A$Rγ2 puncta in both Txf and Nxf (Fig. 4g–l). Consistent with the immunocytochemical data, overexpression of PRR7 reduced both amplitude and frequency of miniature inhibitory postsynaptic currents (mIPSCs) in Txf neurons, but had no effect in Nxf neurons (Fig. 4m–o). Thus, PRR7 did not show the non-cell autonomous effect on GABAergic synapses in neighboring neurons and its loss did not affect inhibitory synapse numbers.

**PRR7 does not induce neuronal cell death.** Previous studies suggest that PRR7 overexpression has pro-apoptotic effects[24,26]. To determine whether PRR7 has apoptosis-inducing activity in hippocampal neurons, we first examined the nuclear morphology

of neurons by 4′,6-diamidino-2-phenylindole staining. After 24 h post transfection, neurons overexpressing PRR7 showed normal dendritic morphology and intact round nuclei indistinguishable from those of Nxf-transfected and EGFP-transfected neurons (Supplementary Fig. 3a, b). Moreover, cell death examined by TO-PRO3 dye exclusion test (non-cell-permeable dye and thus stains only nuclei of dead cells) did not show significant differences in the percentage of dead cells among hippocampal neurons transfected by GFP alone and PRR7-GFP (Supplementary Fig. 3c, d). We next performed immunostaining for an apoptosis marker, cleaved caspase-3[27], together with NeuN to examine whether PRR7 induces apoptosis in neurons. Neither HA-PRR7 nor PRR7-GFP overexpression significantly increased the percentage of cleaved caspase-3-positive neurons compared to GFP overexpression control in both Txf and Nxf groups (Supplementary Fig. 3e, f). In addition, z-DEVD-FMK (cell-permeant selective inhibitor of caspases, blocks apoptosis) failed to block the PRR7 overexpression-induced loss of excitatory synapses (Supplementary Fig. 3g), while it showed its efficacy of increasing pAkt levels (Supplementary Fig. 3h). Moreover, PRR7 overexpression did not

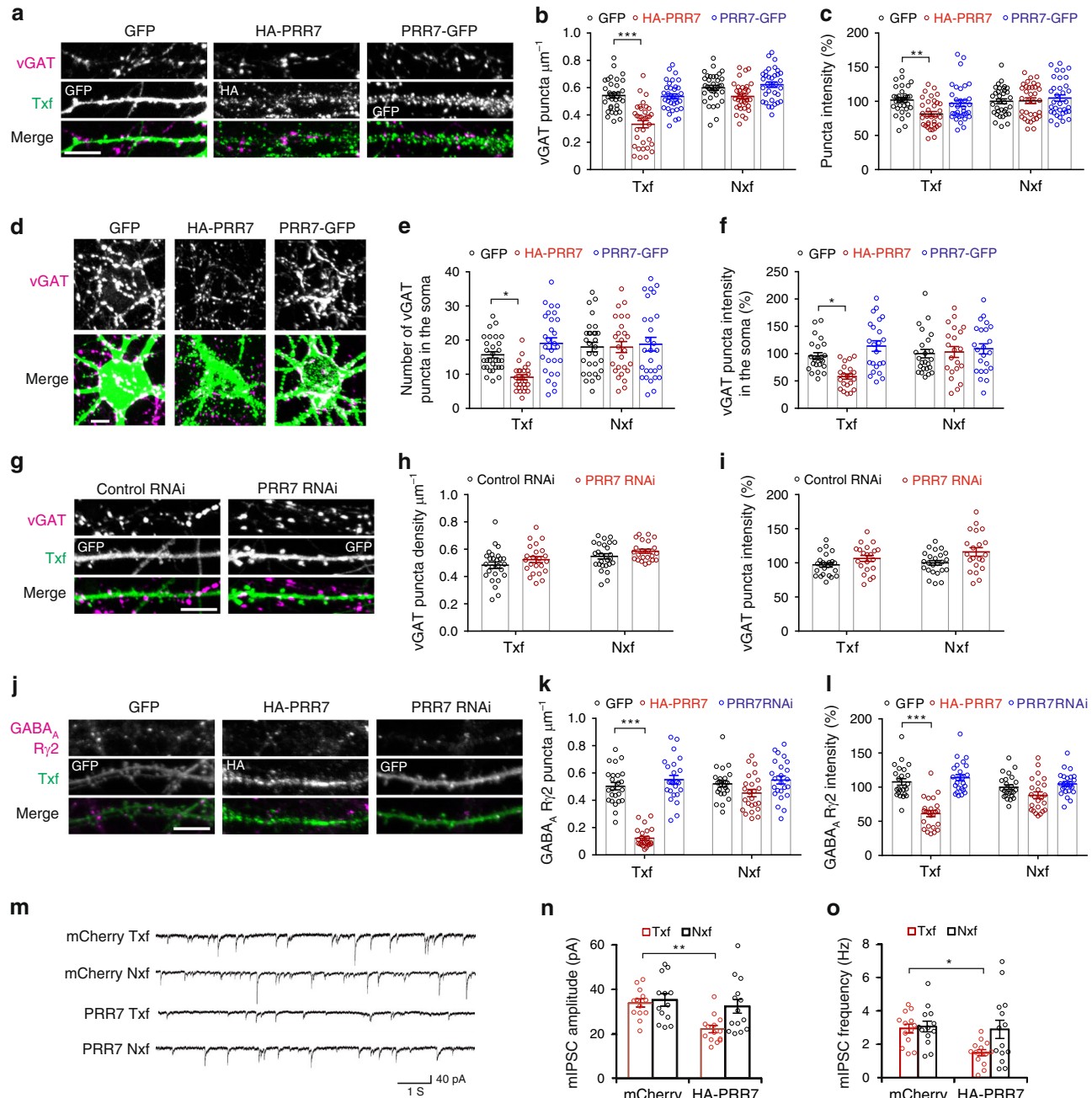

**Fig. 4** Effect of PRR7 on GABAergic synapses. **a** Representative immunofluorescent images of vGAT in the dendrites of hippocampal neurons transfected with GFP, HA-PRR7, or PRR7-GFP. **b**, **c** Quantitation of vGAT puncta densities (**b**) and intensities (**c**) in hippocampal neurons transfected with GFP, HA-PRR7, or PRR7-GFP. $n = 34$ each. **d** Representative immunofluorescent images of vGAT in the soma of hippocampal neurons transfected with GFP, HA-PRR7, or PRR7-GFP. **e**, **f** Quantitation of vGAT puncta densities (**e**) and intensities (**f**) in the soma of hippocampal neurons transfected with GFP, HA-PRR7, or PRR7-GFP. $n = 26$–30. **g** Representative immunofluorescent images of vGAT in the dendrite of hippocampal neurons transfected with Control RNAi or PRR7 RNAi. **h**, **i** Quantitation of vGAT puncta densities (**h**) and intensities (**i**) in hippocampal neurons transfected with control RNAi or PRR7 RNAi. $n = 21$–24. **j** Representative immunofluorescent images of GABA$_A$Rγ2 in the dendrite of hippocampal neurons transfected with control RNAi or PRR7 RNAi. **k**, **l** Quantitation of GABA$_A$Rγ2 puncta densities (**k**) and intensities (**l**) in hippocampal neurons transfected with control RNAi or PRR7 RNAi. $n = 24$ each. **m–o** Effect of PRR7 overexpression on mIPSCs of hippocampal neurons. Representative mIPSC traces (**m**) and quantifications of mIPSC amplitude (**n**) and frequency (**o**). $n = 13$–14. Scale bars, 5 μm. Two-way ANOVA with post hoc Tukey's test: $F_{2,198} = 32.59$ (**c**), $F_{2,198} = 3.963$ (**d**), $F_{2,162} = 6.695$ (**e**), $F_{2,130} = 6.999$ (**f**), $F_{2,198} = 59.26$ (**k**), $F_{2,138} = 37.33$ (**l**), $F_{1,50} = 8.925$ (**n**), $F_{1,50} = 5.01$ (**o**). \*\*\*$P < 0.0001$, \*\*$P = 0.0053$ (**c**), \*$P = 0.025$ (**e**), \*$P = 0.0163$ (**f**), \*\*$P = 0.0072$ (**n**), and \*$P = 0.0328$ (**o**). Data are mean ± SEM

induce the cleavage of Akt, an apoptotic marker (Supplementary Fig. 3h). Taken together, these data suggest that PRR7 overexpression does not induce neuronal apoptosis and that the PRR7-induced synapse loss is not the consequence of apoptotic cell death.

**Exosomal PRR7 removes excitatory synapses.** To study the role of PRR7 secreted on exosomes, we first examined whether PRR7-containing exosomes are sufficient to reduce excitatory synapse numbers. Remarkably, the application of CS harvested from HA-PRR7-transfected neurons (simply called PRR7 CS hereafter) to

naïve neurons for 24 h specifically reduced excitatory synapse numbers (determined by colocalized clusters of PSD-95 and vGLUT1) but not inhibitory synapses (determined by colocalized clusters of GABA$_A$Rγ2 and vGAT) (Fig. 5a, b). On the other hand, control CS from GFP-transfected neurons (GFP CS) or naïve neurons (normal CS) or PRR7 CS devoid of exosomes (PRR7 CS- Exo; prepared by removing exosomes by ultracentrifugation) had no effect on excitatory synapses (Fig. 5a, b and Supplementary Fig. 4). Notably, the application of PRR7-GFP CS had no effect on PSD-95 clusters (Fig. 5c, d), suggesting that PRR7-GFP is defective in exosomal secretion. To examine this possibility, we purified P100 from the CS of neurons transfected with various PRR7 constructs and examined the presence of their proteins in exosomes by western blot analyses (Fig. 5e). Indeed, PRR7-GFP was not detected in P100 exosomes, while the dC4 mutant showed exosomal secretion as effective as HA-PRR7 (Fig. 5e). Therefore, PDZ interaction is not required for the secretion of PRR7 on exosomes, but is necessary for its action in Nxf (as shown by the lack of dC4 effect in Nxf; Fig. 3). We also found that the amount of PRR7 is vastly higher in exosomes purified from PRR7 CS than in those purified from normal CS (Control), indicating that PRR7 overexpression resulted in the greater secretion of PRR7 on exosomes (Fig. 5e). Importantly, replacement of the PRR7 CS with normal conditioned medium at 12 h restored PSD-95 clusters to control levels at 24 h (Fig. 5f), indicating that the effect of PRR7 CS on excitatory synapses is reversible. We next tested whether purified exosomes containing high levels of PRR7 can recapitulate the CS effect on excitatory synapses. The application of P100 exosomes purified from PRR7 CS reduced the number of PSD-95 clusters drastically and rapidly ($t_{1/2}$ <30 min, Fig. 5g and Supplementary Fig. 4a, b). Interestingly, HA-PRR7-containing exosomes purified from HEK293 cells and purified GST-PRR7 proteins failed to replicate the effect of HA-PRR7-containing neuronal exosomes on PSD-95 (Supplementary Fig. 4a, b). Therefore, neuronal exosomes containing higher levels of PRR7 are sufficient to reduce synapse numbers.

To examine whether PRR7-carrying neuronal exosomes are absorbed by neurons, we incubated naïve hippocampal neurons with exosomes purified from PRR7 CS for 2 h and examined for the presence of bound or absorbed exosomal HA-PRR7 by IC. When HA-PRR7-containing exosomes were spot applied directly, strong HA immunofluorescent staining was observed in the dendrites and axons of recipient neurons (Fig. 5h). In contrast, control exosomes purified from the CS of GFP-transfected neurons did not show such staining patterns. When HA-PRR7-containing exosomes were applied to neurons after thorough mixing with the conditioned medium, total staining of neurons (with membrane permeabilization) showed weak but specific immunofluorescence in the somatodendritic areas, which was not observed in surface staining (Fig. 5i). Bound or absorbed HA-PRR7 was also readily detected by western blotting analyses of the total protein extracts prepared from the exosome-treated neurons after extensive washing (Fig. 5j). These results collectively suggest that PRR7-containing exosomes fuse with the PM of recipient neurons to release cargo PRR7.

**Exosomal PRR7 is required for its non-cell autonomous action.** Although exosomes carrying high levels of PRR7 showed the ability to eliminate excitatory synapses, the data do not fully establish that the non-cell autonomous effect of PRR7 overexpression is solely mediated by mechanisms dependent on exosomes. Therefore, we next investigated whether the exosomal secretion of PRR7 is required for synapse elimination in neighboring neurons. First, we examined PRR7 trafficking in hippocampal neurons. Since exosomes originate from intraluminal

vesicles in MVBs, it is expected that newly synthesized PRR7 is first exocytosed to the PM and then undergoes endocytic trafficking that eventually leads to exosomes via MVB–PM fusion (Supplementary Fig. 5a). Consistent with the trafficking model, HA-PRR7 showed partial colocalization with early endosome marker EEA1 in dendrites and soma (Supplementary Fig. 5b). At 3 h post transfection, HA-PRR7 showed tubulovesicular patterns in the soma, which are highly overlapping with late endosome marker Rab9 and partially colocalized with late endosome/lysosome marker Lamp1 (Supplementary Fig. 5b). Notably, at the early time point, while exosome-localized PRR7 was clearly found, PRR7 showed little colocalization with PSD-95 in dendrites (Supplementary Fig. 5c), suggesting that PRR7 does not first traffic to synapses before being secreted on exosomes.

We next thought to block the exosomal secretion of PRR7. Different species of Rab proteins are reported to be involved in exosome biogenesis and secretion in various cell types[22,28,29], which include Rab27b and Rab11a that are highly expressed in the brain. Therefore, we examined the effect of inhibiting these Rab functions on the exosome secretion of PRR7 using dominant-negative (DN) constructs[30]. Remarkably, overexpression of CFP-tagged DN form of Rab27b (T23N), but not Rab27b WT, completely blocked the exosomal secretion of co-transfected HA-PRR7 (Fig. 6a). However, the total amount of exosomes in CS (determined by Flotillin-1) was not affected much by the co-transfection of Rab27b DN, reflecting sparse transfection of neurons (<0.1%). Overexpression of Rab27b DN also blocked the HA-PRR7-induced elimination of PSD-95 clusters specifically in Nxf, while Rab27b WT, Rab11a WT, or Rab11a DN (S25N) did not (Fig. 6b–d). Therefore, these data suggest that exosomal secretion of PRR7 is required for its effect in Nxf and Rab27b, not Rab11, is involved in the exosomal secretion of PRR7 by neurons. To further corroborate the findings, we also examined the effect of APV which hampered the exosomal secretion of PRR7 (Fig. 1f). Incubation of neurons with APV or CNQX for 6 h did not significantly changed PSD-95 clusters in GFP control (Fig. 6f). In contrast, APV completely abolished PSD-95 cluster loss by HA-PRR7 in Nxf but CNQX did not (Fig. 6e, f). Thus, exosomal PRR7 is necessary and sufficient for the elimination of excitatory synapses in neighboring neurons.

**PRR7 promotes proteasomal degradation of PSD-95.** We next investigated molecular mechanisms by which PRR7 removes excitatory synapses. Ubiquitin–proteasome system (UPS)-mediated protein degradation is one of the major mechanisms for synapse modification and elimination[31,32]. Since PRR7 eliminates PSD-95 clusters relatively fast (Fig. 5g), we first thought of the possibility that PRR7 induces the UPS-dependent degradation of PSD proteins. Incubation of hippocampal cultured neurons with proteasome inhibitors alone (lactacystin or MG132) did not change PSD-95 cluster numbers significantly[33]. In contrast, proteasome inhibitors blocked the PRR7 overexpression-induced loss of PSD-95 clusters in the dendrites of Nxf (Fig. 7a, b). Moreover, the proteasome inhibitors promoted the accumulation of PSD-95 aggregates in the soma of HA-PRR7 Txf (Fig. 7a), which is frequently observed for poly-ubiquitinated proteins[33]. These data suggest that PRR7 induces the degradation of PSD-95. Corroborating the immunocytochemical data, PRR7 overexpression reduced total PSD-95 protein levels and lactacystin prevented the reduction (Fig. 7c, d). Other synaptic scaffolding proteins, MAGUKs and SAPAPs, also showed similar changes to PSD-95 (Supplementary Fig. 6). Furthermore, PRR7 overexpression also increased the amount of total poly-ubiquitinated proteins as determined by antibodies recognizing K48-specific poly-ubiquitination (Fig. 7c, d and Supplementary Fig. 6).

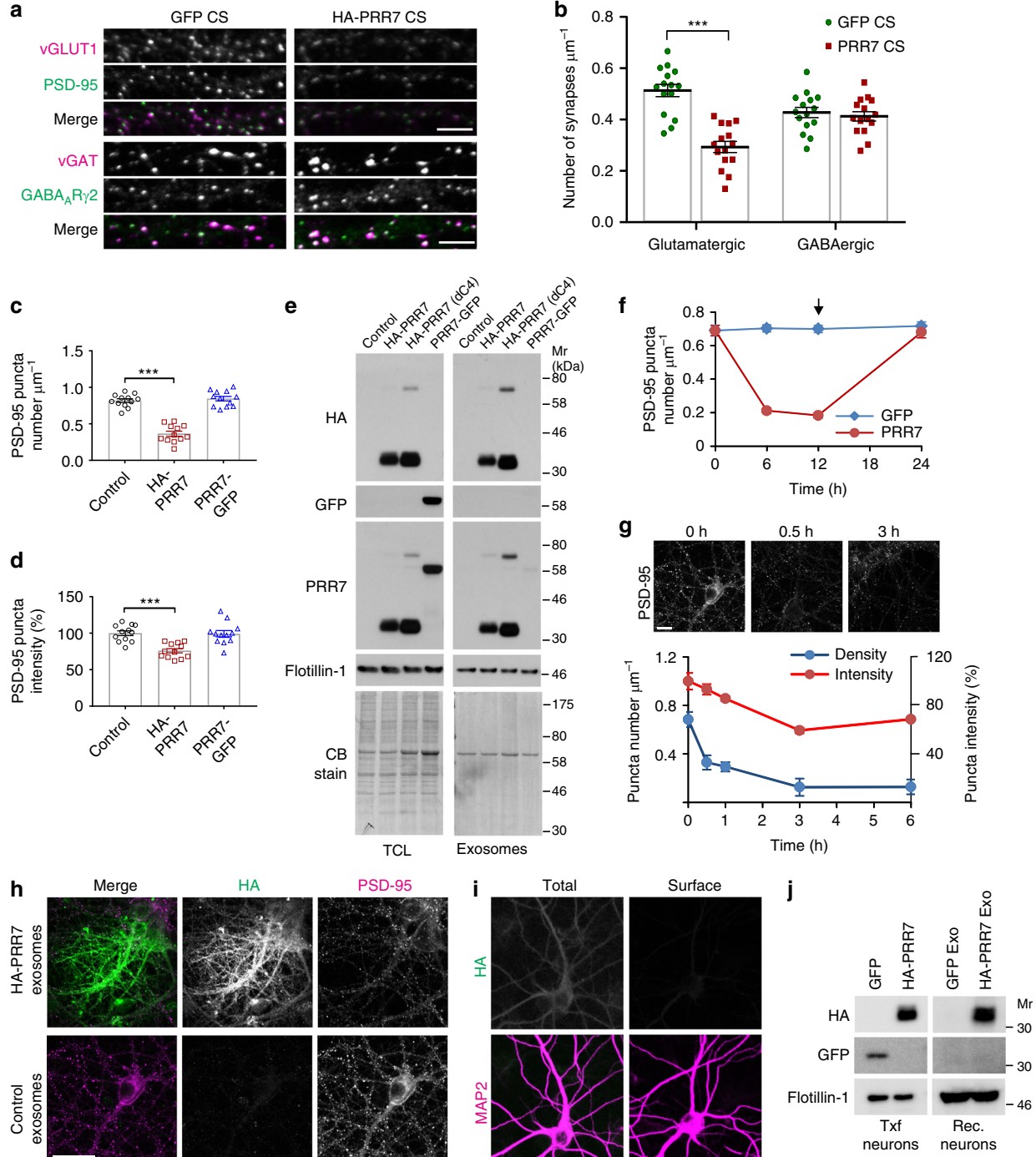

**Fig. 5** Exosomal PRR7 is sufficient for the removal of excitatory synapses. **a** Representative immunofluorescent images of PSD-95/vGLUT and vGAT/ GABA$_A$R$\gamma$2 in hippocampal neurons treated with CS harvested from GFP control-transfected or HA-PRR7-transfected neurons. **b** Quantitation of the CS effect on excitatory and inhibitory synapses. **c**, **d** Quantitation of the CS effects on PSD-95 harvested from GFP-transfected, HA-PRR7-transfected or PRR7-GFP-transfected neurons. Cluster density (**c**) and intensity (**d**). $n = 12$ per condition. **e** Exosomal secretion of various PRR7 mutants. CB Coomassie blue staining. **f** Reversal of PSD-95 clusters by replacing PRR7-overexpressing CS with normal conditioned medium at 12 h. **g** Time course of the effect of HA-PRR7-exosomes on PSD-95. Top panels show representative images and quantified data are shown bottom. $n = 15$ each. **h**–**j** Uptake of exosomal PRR7 by neurons. Representative images of hippocampal neurons treated with exosomes purified from either GFP-transfected (control) or HA-PRR7-transfected neuron culture. Total staining of HA and PSD-95 in recipient neurons after the spot application of exosomes (**h**). Total and surface staining of HA and MAP2 after the application of evenly mixed HA-PRR7-containing exosomes (**i**). Western blotting analyses of TCLs prepared from Txf and exosome-treated recipient (Rec.) neurons (**j**). Scale bars, 5 (**a**), 10 (**g**), and 20 μm (**h**, **i**). All data are mean ± SEM. Two-way (**b**) or one-way ANOVA (**c**, **d**) Post hoc Tukey's test, $F_{1,56} = 30.71$ (**b**), $F_{2,33} = 78.78$ (**c**), and $F_{2,33} = 14.66$ (**d**). ***$P < 0.0001$. Mr relative molecular weight

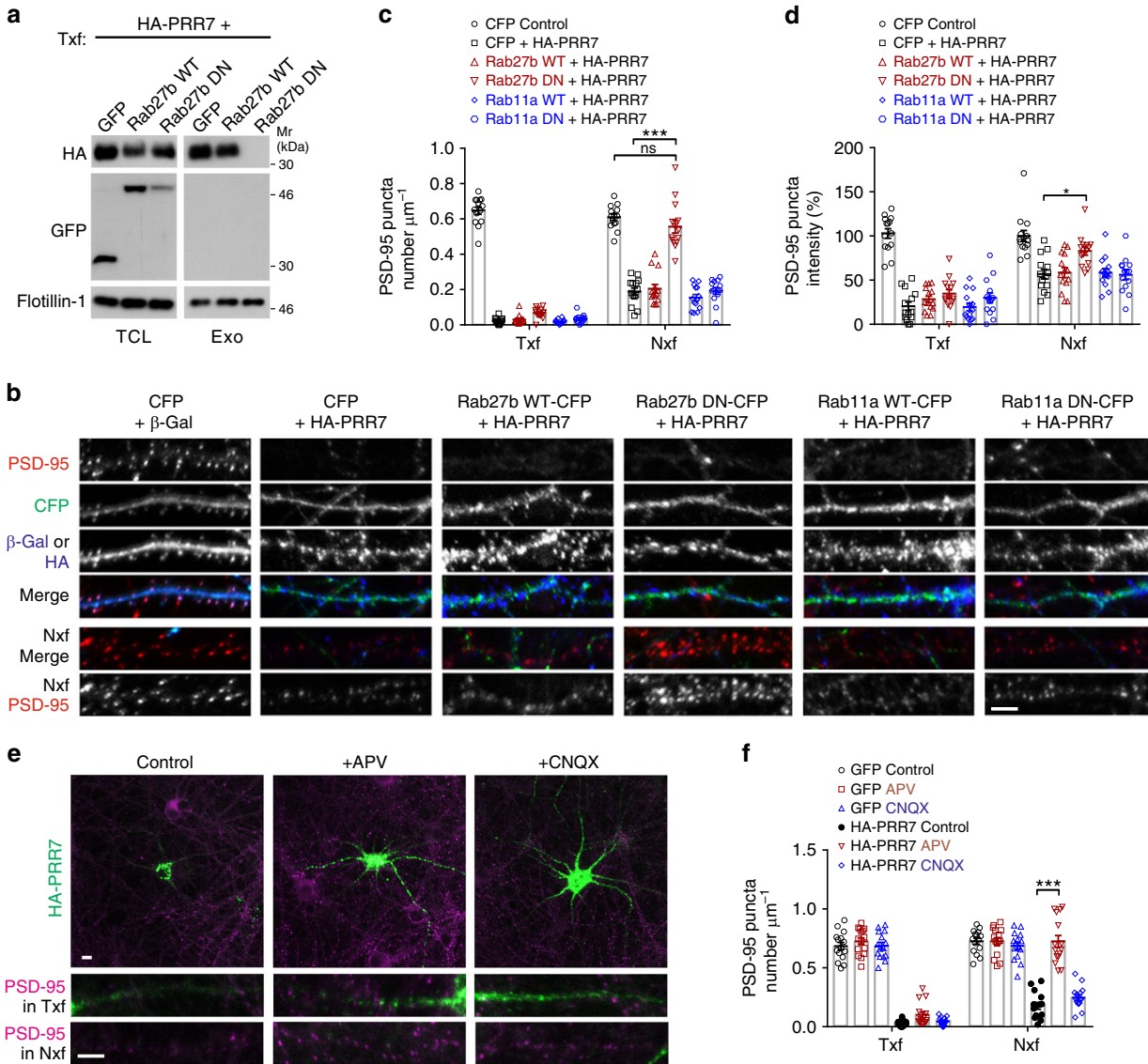

**Fig. 6** Exosomal secretion of PRR7 is required for the elimination of excitatory synapses in neighboring neurons. **a** Blockade of exosomal secretion of HA-PRR7 by Rab27b DN. **b–d** Effect of Rab11a and Rab27 WT and DN on the PRR7-induced loss of PSD-95 clusters. Representative immunofluorescent images of PSD-95 in hippocampal neurons transfected with the constructs indicated above images (**b**). Quantitation of the data on PSD-95 cluster density (**c**) and intensity (**d**). $n = 14$ per condition. **e, f** Effect of APV and CNQX on the PRR7-induced loss of PSD-95 clusters. Representative immunofluorescent images of PSD-95 (**e**) and quantitation of the data (**f**). $n = 15$ each. Scale bars, 5 μm. Two-way ANOVA, post hoc Tukey's test: $F_{5,156} = 278.4$ (**c**), $F_{5,156} = 45.49$ (**d**), $F_{5,168} = 199.7$ (**f**). ***$P < 0.0001$ and *$P = 0.0197$. Data are mean ± SEM. Mr relative molecular weight

Analyses of PSD-95 immunoprecipitated under denaturing conditions (to minimize co-immunoprecipitation) verified that PRR7 overexpression increases poly-ubiquitinated PSD-95 levels (146 ± 4.5%, $n = 3$; Fig. 7e). Therefore, PRR7 promotes the degradation of PSD-95 and other PSD scaffolding proteins by the UPS.

**PRR7 inhibits synaptogenic Wnt signaling.** The strong non-cell autonomous effect of PRR7 RNAi on Nxf was rather surprising (Fig. 2f, h) because the total amount of endogenous PRR7 was not affected much due to the low transfection efficiency (Fig. 8e). Intriguingly, synaptogenic Wnt5a and Wnt7a are also secreted on exosomes (Fig. 1c). Therefore, we speculated that PRR7 might inhibit the synaptogenic Wnt signaling. To address this possibility, we first tested if the augmentation of Wnt signaling can rescue the synapse loss induced by PRR7 overexpression. Strikingly, bath application of recombinant Wnt5a or Wnt7a restored

PSD-95 clusters in Nxf of PRR7-transfected neuron culture (Fig. 8a, b). Conversely, addition of soluble Wnt inhibitors (sFz5-Fc or sFRP1) to the culture medium completely abolished the effect of PRR7 knockdown on PSD-95 in both Txf and Nxf (Fig. 8c, d). Thus, the modulation of Wnt signaling is sufficient to negate both the PRR7 overexpression and knockdown effect on excitatory synapses.

To examine whether PRR7 interferes with Wnt signaling rather than acts on unrelated parallel pathways, we examined the effect of PRR7 on the key downstream components of Wnt signaling. Activation of $Ca^{2+}$/CaM-dependent protein kinase II (CaMKII) is a key component of non-canonical Wnt signaling pathway involved in the regulation of excitatory synapses[10,12]. Remarkably, PRR7 overexpression decreased the amounts of active auto-phosphorylated CaMKIIα and β, while PRR7 knockdown increased the amounts of auto-phosphorylated CaMKIIα and β (Fig. 8e, f). Thus, PRR7 inhibits the non-canonical CaMKII-

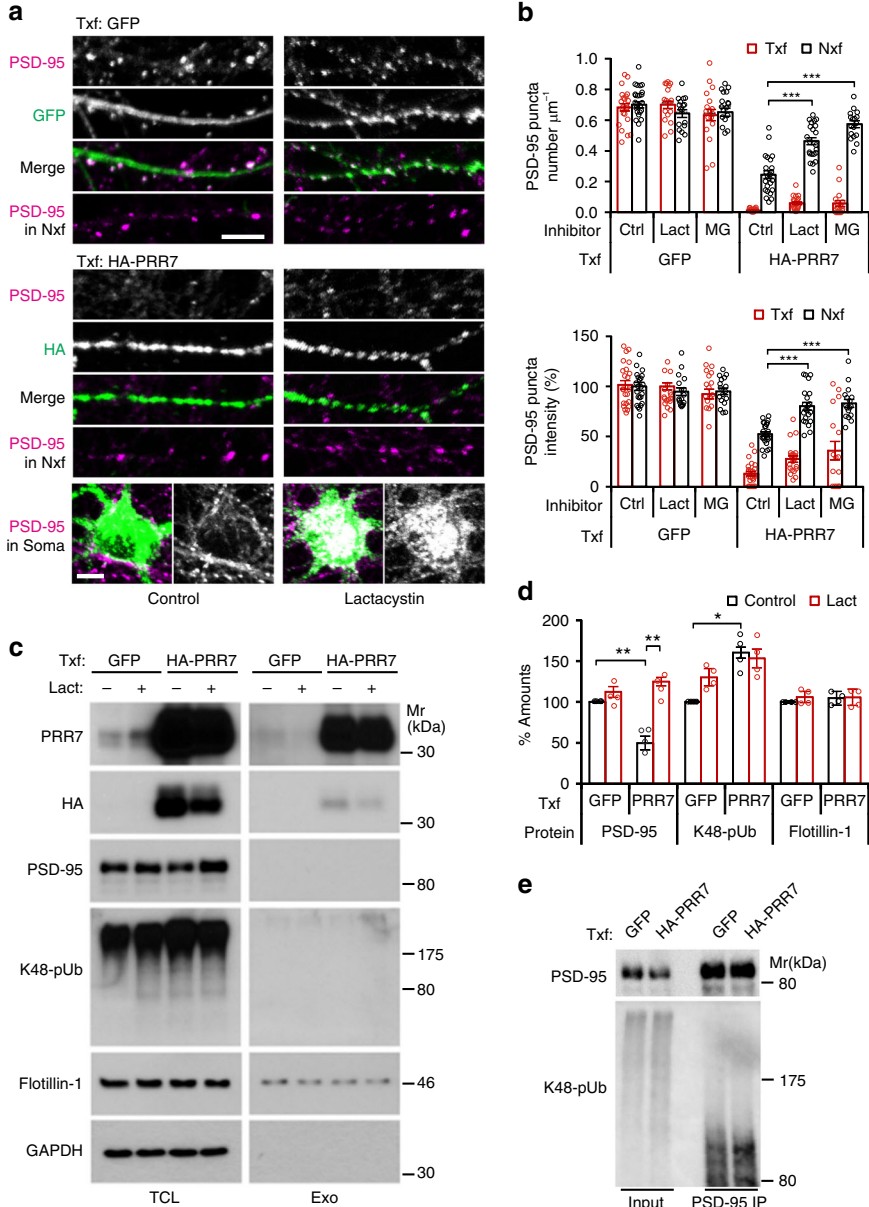

**Fig. 7** PRR7 promotes the proteasomal degradation of synaptic proteins. **a, b** Effect of proteasome inhibitors on PRR7-induced removal of PSD-95 clusters. Representative immunofluorescent images of PSD-95 in the dendrites and soma (Lactacystin-treated) from GFP-transfected or HA-PRR7-transfected neurons (**a**), and quantitation of data (**b**). Scale bars, 5 μm. Lact lactacystin, MG MG132-treated neurons. $n = 17$–$24$. **c, d** Effect of PRR7 overexpression on the protein levels of major PSD scaffold proteins and poly-ubiquitination (pUb), in the absence (control) or presence of Lact. Representative immunoblots of TCL and purified exosomes from the CS (**c**) and quantified data for the protein levels in TCL (**d**). $n = 4$. **e** Increased poly-ubiquitination of PSD-95 by PRR7 overexpression. IP immunoprecipitation, Mr relative molecular weights. Two-way ANOVA, post hoc Tukey's test: $F_{5,230} = 222.3$ (**b**, density), $F_{5,230} = 98.42$ (**b**, intensity), $F_{1,12} = 7.098$ (**d**, PSD-95), and $F_{1,12} = 23.87$ (**d**, K48-pUb). ***$P < 0.0001$, **$P < 0.01$, and *$P < 0.05$. Data are mean ± SEM

mediated synaptogenic Wnt signaling. Moreover, PRR7 also affected canonical Wnt signaling by glycogen synthase kinase-3β (GSK3β). PRR7 overexpression led to the increased activity of GSK3β as shown by the increased amount of tyrosine-phosphorylated GSK3β (pY-GSK3β). Conversely, PRR7 knock-down increased the amount of the inactive form of GSK3β (pS-GSK3β), indicating the inhibition of GSK3β. Total amounts of GSK3β, CaMKIIα, and CaMKIIβ were not significantly affected by PRR7 overexpression or knockdown. These data suggest that PRR7 inhibits Wnt signaling.

Since Wnt5a and Wnt7a are also secreted on exosomes (Fig. 1c), we next thought of the possibility that PRR7 affects Wnt secretion. Strikingly, in hippocampal neurons, PRR7 knockdown increased the amounts of secreted Wnt5a and Wnt7a on exosomes (Fig. 8g), while PRR7 overexpression decreased the amount of Wnt7a in exosomes (Supplementary Fig. 7a). When expressed in HEK293 cells, PRR7 strongly reduced the surface expression of Wnt7a (Supplementary Fig. 7b). Interestingly, PRR7 dramatically altered the subcellular localization of GPR177, the mouse ortholog of Evi/Wls important for Wnt secretion[18,34]. When expressed in neurons, myc-GPR177 showed strong localization in the dendritic spines (Supplementary Fig. 7c). In contrast, when co-expressed with PRR7, myc-GPR177 is mostly confined to vesicular structures in the soma which colocalized with PRR7 (Supplementary Fig. 7c). These data strongly suggest that PRR7 inhibits the secretion of Wnts, which explains the

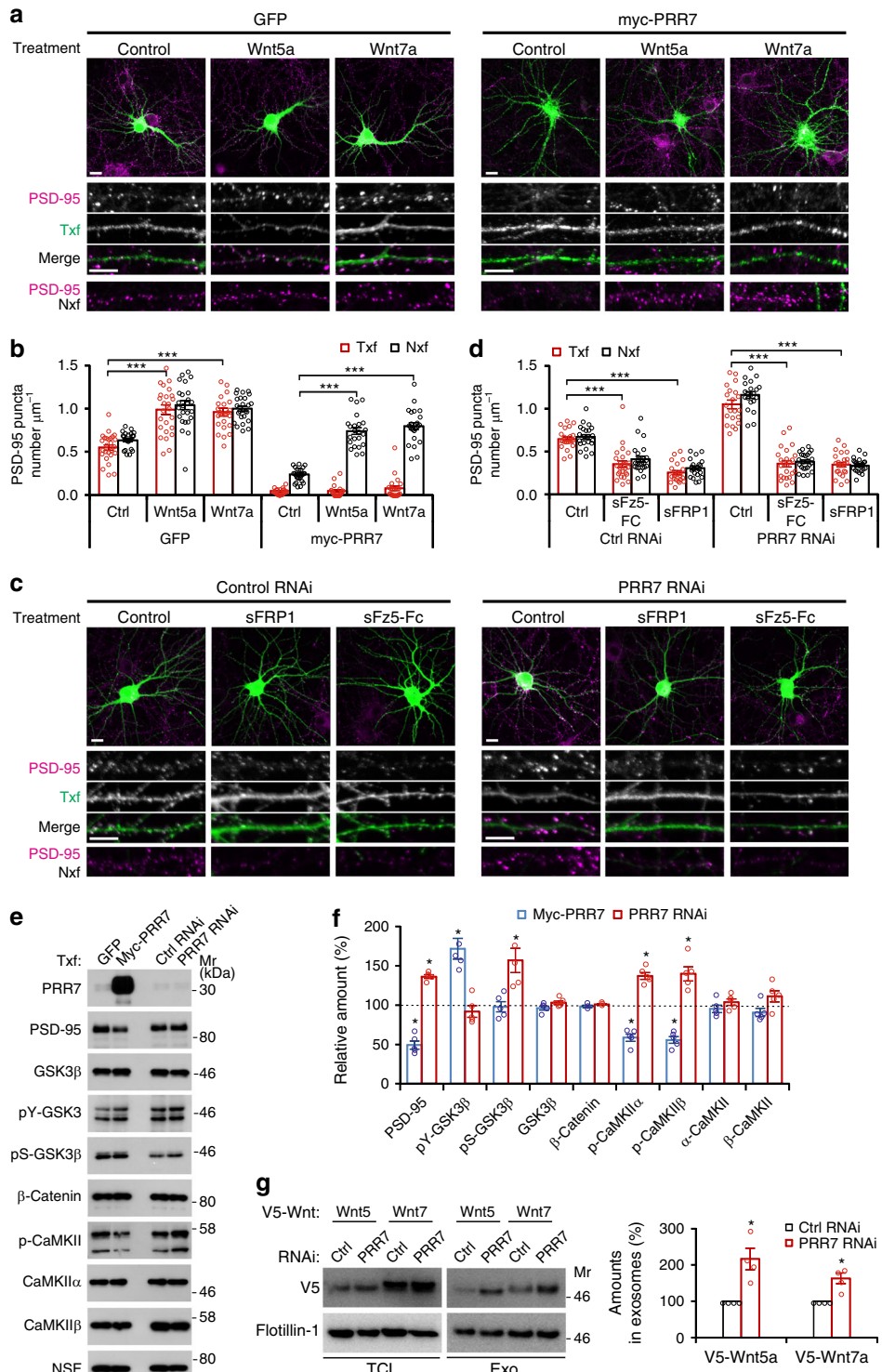

**Fig. 8** PRR7 inhibits synaptogenic Wnt signaling. **a**, **b** Effect of exogenous application of Wnt5a and Wnt7a on PRR7-induced loss of PSD-95 clusters. Representative immunofluorescent images of PSD-95 (**a**) and quantitation of data (**b**). $n = 23$–$26$. **c**, **d** Effect of soluble Wnt inhibitors on PRR7 knockdown-induced increase of PSD-95 clusters. Representative immunofluorescent images of PSD-95 (**c**) and quantitation of data (**d**). $n = 22$–$24$. **e**, **f** Modulation of Wnt signaling by PRR7. Representative immunoblots showing the effect of PRR7 overexpression or knockdown on the indicated Wnt signaling components (**e**) and quantification of the data (**f**). $n = 3$–$5$. **g** Effect of PRR7 knockdown on the exosomal secretion of Wnt5a and Wnt7a. $n = 4$. Data are mean ± SEM. Scale bars, 5 µm. Two-way ANOVA (**b**, **d**), post hoc Tukey's test: $F_{2,148} = 66.62$ (**b**, GFP), $F_{2,138} = 76.96$ (**b**, myc-PRR7), $F_{2,130} = 86.74$ (**d**, Ctrl RNAi), and $F_{2,134} = 336.6$ (**d**, PRR7 RNAi). ***$P < 0.0001$; Student $t$ test (**f**, **g**) and *$P < 0.05$. Mr relative molecular weight

substantial impact of PRR7 knockdown on the excitatory synapses in Nxf and the blockade of the knockdown effect by soluble Wnt inhibitors (Fig. 8d). These data also raised the interesting possibility that Wnts might also affect the exosomal secretion of PRR7. Remarkably, mimicking Wnt-induced GSK3β inhibition conditions by GSK3 inhibitor (LiCl) or GSK3β knockdown reduced the amount of PRR7 secreted on exosomes (Supplementary Fig. 8a, b). Moreover, pharmacological inhibition

(LiCl or TDZD-8) or knockdown of GSK3β also prevented the PRR7 overexpression-induced loss of PSD-95 in Nxf (Supplementary Fig. 8c, d). Therefore, these data suggest that PRR7 and Wnts reciprocally inhibit their exosome secretion.

To corroborate the in vitro data, we also examined the effect of PRR7 gene deletion in vivo on excitatory synapses. PRR7 knockout (KO) mice showed a drastic increase in the protein levels of key components of excitatory synapses including PSD-95 MAGUKs, SAPAPs, and ionotropic glutamate receptors, without discernable changes in the amounts of GABAergic synapse components (Supplementary Fig. 9a). Furthermore, PRR7 KO mice displayed reduced GSK3β activity and increased CaMKII activity (Supplementary Fig. 9a). Interestingly, the amount of active β-catenin is also increased in the KO mice, suggesting enhanced canonical Wnt signaling. Lastly, reduced levels of PRR7 in heterozygote PRR7 KO mice were sufficient to increase the numbers of vGLUT1 puncta in the cortex without affecting vGAT puncta numbers (Supplementary Fig. 9b). Moreover, complete PRR7 loss of function in homozygous PRR7 KO mice significantly increased vGLUT1 puncta density in the hippocampus (Supplementary Fig. 9c). These in vivo data are consistent with the specific increase in excitatory synapse numbers by the loss of PRR7 function. Taken together, these data strongly suggest that PRR7 acts as a physiological Wnt inhibitor for synapse regulation.

## Discussion
In this study, we showed that excitatory synapse numbers in neurons are controlled by the mutually opposing actions of Wnts and PRR7, which promote synaptogenesis and elimination, respectively. Moreover, we found that exosomes play an unexpected role as the signal carrier in this processes.

Accumulating evidence indicates that Wnts play roles in mature neurons and adult brains[6,7]. For example, local application of sFRP1 caused a drastic reduction of synapse numbers in CA3 neurons of the adult hippocampus[8] and bath incubation of Dkk1 induced a rapid loss of excitatory synapses in mature cultured hippocampal neurons[15]. Based on these findings, Wnt signaling is thought to be required for the maintenance of synapses via unknown molecular mechanisms. Considering the surprisingly rapid effect of Wnt inhibitors (0.5 h)[15], much faster than the normal turnover of synaptic proteins (1 day)[35], and remarkably stable synapse numbers in mature neurons actively expressing synaptogenic Wnts[5], it is reasonable to posit that neurons also express synapse-eliminating factors. We propose that PRR7 is a secreted factor functioning as the synapse-removal factor and balances the pressure of Wnt-driven synaptogenesis under normal conditions to maintain synapse numbers. This reciprocal mechanism would also allow neurons to rapidly adjust synapse numbers upon strong plasticity-inducing activity by simply shifting the balance of secreted Wnts and PRR7 amounts. The high expression of PRR7 in healthy adult brains additionally supports the proposed function of PRR7.

Our data suggest that exosomal secretion of PRR7 is NMDAR-dependent. Notably, enhancing activity by Bic did not further increase the amount of PRR7 secreted on exosomes. Therefore, basal level of activity seems sufficient for the exosomal secretion of PRR7. Previous studies indicate that Wnts are also activity-dependently released[8,9,36]. However, Wnts secretion might require stronger NMDAR activation like tetanic stimulations[9]. Additionally, the amount of released Wnts might be dependent on their relative protein levels dictated by activity-dependent transcription[36].

Since the first demonstration of exosome secretion by central neurons more than 10 years ago[19], a few neuronal proteins have been shown to be secreted on exosomes including AMPA receptors (AMPARs), amyloid β, ephrins, and Arc[20,37–40]. However,

the physiological significance of these exosome-associated neuronal proteins in the vertebrate brain is poorly understood[22]. Here we demonstrated that exosomal PRR7 is necessary and sufficient for synapse elimination in local neurons not only in dissociated hippocampal neuron culture but also in organotypic hippocampal slice culture. Moreover, we also showed that the extracellular release of PRR7 on exosomes is a regulated process controlled by neuronal activity and GSK3β activity. Importantly, unlike exosomal PRR7, the amount of total released exosomes remains relatively stable, suggesting that neurons can change cargo molecules that are shipped on exosomes. Therefore, these data provide the most substantial experimental evidence, to the best of our knowledge, that exosomes play a significant role in inter-neuronal communication.

Notably, exosomal PRR7 secreted by neurons is uptaken almost exclusively by neurons. This is consistent with the report showing that neuronal exosomes are specifically bound by neurons and not by glia cells[41]. Interestingly, PRR7-containing neuronal exosomes seem to fuse with the PM of neurons to release the cargo PRR7. This is reminiscent of the PM fusion of dendritic cell-derived exosomes in the immune system[42], and supports the functionality of exosomes in the intercellular transfer of signaling molecules directly to the PM and cytoplasm of recipient cells. Although exosomes contain many protein species and miRNAs, it is highly likely that exosomal PRR7 is the main player promoting synapse elimination, considering that PRR7-GFP is not packed into exosomes but retains the capacity to reduce synapse numbers in Txf neurons. Nonetheless, it remains a possibility that PRR7 promotes the co-transport of unidentified specific molecules (RNAs or proteins) in exosomes to exert its effect on neighboring neurons.

How does PRR7 promote synapse removal? Our studies indicate that PRR7 exerts its effect via multiple novel mechanisms distinguished from those of conventional Wnt inhibitors. The common features of Wnt inhibitors are to antagonize Wnt signaling by preventing ligand–receptor interaction or Wnt receptor maturation[14]. In contrast, our data suggest that PRR7 inhibits Wnts by blocking their secretion and also by activating GSK3β. In addition, PRR7 promotes the proteasomal degradation of synaptic scaffolding proteins, in contrast to Wnts that inhibit the degradation of β-catenin. The activation of GSK3β is one of the important characteristics of PRR7 to consider it as a Wnt inhibitor since GSK3β inhibition is the central component of Wnt signaling. Importantly, GSK3β activation also promotes the removal of PSD-95 and AMPA receptor from synapses[43,44], which likely occurs downstream of PRR7 in the synapse elimination process. The detailed molecular mechanisms by which PRR7 inhibits Wnt secretion, activates GSK3β, and promotes PSD protein degradation remain to be determined in the future. In our studies, exogenous application of Wnts rescues the PRR7 overexpression phenotype in Nxf. Therefore, it is unlikely that PRR7 interferes with the surface expression of Fz receptors, which is the action mechanism proposed for the zebrafish ortholog Ottogi[45]. Instead, our data suggest that PRR7 inhibits Wnt secretion by potentially interfering with GPR177 function[46]. Considering that PRR7 reduces Wnt secretion, the increased activation of GSK3β might be partly due to decreased Wnt signaling. Unlike the non-cell autonomous mechanism, the effect of PRR7 in Txf neurons was not blocked by protease inhibitors, Wnts, or GSK3β inhibition. Therefore, the cell autonomous effect might be mediated by unidentified additional mechanisms. Since PRR7 can enter the nucleus[24], it is a possibility that PRR7 might regulate the transcriptional activity of neurons and interfere with the transcription of proteins necessary for synapse maintenance.

In contrast to excitatory synapses, PRR7 overexpression showed moderate effect on GABA synapses only in a cell

autonomous manner. Since GSK3β promotes the degradation of gephyrin and reduces GABA$_A$R clustering[47,48], the activation of GSK3β by PRR7 overexpression is likely responsible for the GABA synapse loss. This is consistent with the findings that Wnt5a promotes recycling of functional GABA$_A$Rs via CaMKII that inhibits GSK3β[49,50]. Alternatively, the reduction of inhibitory synapses might be due to homeostatic changes induced by major loss of excitatory synapses in PRR7-transfected neurons[51]. However, knockdown or KO of PRR7 had no influence on GABAergic synapse numbers. Moreover, the application of sFRP did not reduce GABAergic synapse numbers in cultured hippocampal neurons[49] and blockade of the canonical Wnt signaling by inducible Dkk1 expression did not affect hippocampal GABAergic synapses in vivo[7]. Therefore, it remains to be further investigated whether PRR7 and Wnts control GABAergic synapses under physiological conditions.

In our experimental conditions, PRR7 removes synapses independently of neuronal cell death. This contrasts to the previous studies reporting that PRR7 overexpression promotes neuronal death but has no effect on NMDAR or AMPAR currents[24]. At present, the bases of the discrepancy are unclear but are most likely due to differences in experimental methods and conditions: for example, the duration of PRR7 overexpression (<1 day vs. 2–7 days) and differences in neuron culture (might contain different populations of neurons and glial cells). Importantly, the effects of PRR7 on excitatory synapse numbers we described is physiologically relevant since PRR7 KO mice showed increased excitatory synapse numbers.

Finally, we demonstrated that PRR7 overexpression or knockdown in a CA1 pyramidal cell has a large influence on the synapses in directly neighboring neurons. This is remarkable because these neurons typically do not directly form synapses. Therefore, the exosomal PRR7-mediated synaptic plasticity seems to operate in a manner dependent on proximity regardless of synaptic connection. Further studies on how far exosomes carrying PRR7 travel in the brain will reveal the scope of the exosome-mediated inter-neuronal signaling.

## Methods

**Animals.** PRR7 KO mice (Prr7$^{tm1(KOMP)Vlcg}$; Project ID VG10972) were obtained from KOMP Repository (UC Davis). Both sexes of mice were used for the analyses. Genotyping was performed using the following sets of primers: TUF (5′-GCACC TACACGTTCCTCACTTG-3′) and TUR (5′GCAGAGCAGGACGATGAGA C-3′); SU (5′-TCGCTGAGCCCTGTGTCTG-3′) and LacInZRev (5′-GTCTGTCC TAGCTTCCTCACTG-3′). Timed-pregnant female Sprague–Dawley rats were obtained from Envigo. All experimental procedures involving the animals were approved by the Institutional Animal Care and Use Committee in the Medical College of Wisconsin.

**Primary cultured hippocampal neurons.** Dissociated hippocampal neuron cultures were prepared from E18 to E19 embryos of Sprague–Dawley rats and maintained in Neurobasal media supplemented with B27 and glutamine[33]. For IC and electrophysiology, hippocampal neurons were plated onto glass coverslips (Carolina Biologicals) pre-coated with poly-D-lysine (25 µg ml$^{-1}$; Corning) and laminin (2.5 µg ml$^{-1}$; Corning) at the density of $7.5 \times 10^4$ cells per well in 12-well tissue culture plates. For large-scale exosome preparation, rat cortical neurons were plated at the density of $1 \times 10^6$ cells on standard 100-mm-diameter tissue culture dishes pre-coated with poly-D-lysine and laminin.

**Organotypic hippocampal slice culture.** Organotypic slice cultures were prepared from the hippocampi of 7-day-old Sprague–Dawley rats. Slices (400 µm thickness) were prepared using a tissue chopper, cultured in medium (minimum essential medium supplemented with 1 µg ml$^{-1}$ insulin, 0.0012% ascorbic acid, 20% horse serum, 1 mM L-glutamine, 1 mM CaCl$_2$, 13 mM D-glucose, 5.2 mM NaHCO$_3$, 30 mM HEPES, penicillin–streptomycin) on top of MilliCell culture plate inserts (Millipore) for 4–6 days. Media were changed every other day.

**Cell lines.** HEK293 (CRL-1573) cells were obtained from the ATCC and grown in Dulbecco's modified Eagle's medium supplemented with 10% bovine serum and penicillin–streptomycin (ThermoFisher Scientific) at 37 °C, 5% CO$_2$. For surface

staining experiments, cells were grown on glass coverslips pre-coated with poly-L-lysine (0.5 mg ml$^{-1}$; Sigma) dissolved in dH$_2$O.

**Antibodies, reagents, and inhibitors.** Primary antibodies and their dilution factors used for IC, immunohistochemistry (IHC), and western blotting (W) are: Akt (CST, Cat#: 4691, 1:1000 W), phospho-Akt (CST, Cat#: 13038, 1:1000 W), Alix1 (BD, Cat#: 611620, 1:1000 W), β-Gal (Promega, Cat#:Z378A,1:1000 IC), active β-catenin (Millipore, Cat#: 05-665, 1:500 W), α-CaMKII (ThermoFisher Scientific, Cat#: 13-7300, 1:1000 W), β-CaMKII (ThermoFisher Scientific, Cat#: 13-9800, 1:1000 W), phospho-CaMKII (Phospho Solutions, Cat#: p1005-286, 1:1000 W), cleaved caspase-3 (CST, Cat#: 9661, 1:1000 W, 1:400 IC), EEA1 (BD, Cat#: 610456, 1:250 IC), Flotillin-1 (BD, Cat#: 610820, 1:2000 W), GABA$_A$Rα2 (Synaptic Systems, Cat#: 224102, 1:250 IC; 1:1000 W), GABA$_A$Rγ2 (Synaptic systems, Cat#: 224003, 1:250 IC; 1:1000 W), GAPDH (CST, Cat#: 2118, 1:1000 W), gephyrin (Synaptic Systems, Cat#: 147111, 1:1000 W), GFAP (Millipore, Cat#: AB5804, 1:1000 IC), GFP (Abcam, Cat#: ab290, 1:10,000 IC; 1:5000 W), GluR2/3 (Millipore, Cat#: AB1506, 1:1000 W), GSK3β (CST, Cat#: 9832, 1:1000 W), phospho-GSK3β, S9 (CST, Cat#: 5558, 1:1000 W), phospho-GSK3, Y279/Y216 (Millipore, Cat#: 05-413, 1:1000 W), rat-HA (Roche, Cat#: 11867423001, 1:200 IC), rabbit-HA (CST, Cat#: 3724, 1:200 IC), Lamp2 (ThermoFisher Scientific, Cat#: 51-2200, 1:500 W; 1:400 IC), mouse-myc (Santa Cruz, Cat#: sc-40, 1:100 IC), rabbit-myc (CST, Cat#: 3724, 1:200 IC), NeuN (Millipore, Cat#: MAB377, 1:75 IC), NSF (Millipore, Cat#: NB21, 1:2000 W), PRR7 (Abnova, Cat#: MAB6689, 1:150 IC; 1:1000 W), PSD-95 (NeuroMab, Cat#: 75-028, 1:400 IC; 1:5000 W), pan-MAGUK (NeuroMab, Cat#: 73-029, 1:400 IC; 1:1000 W), pan-SAPAP (NeuroMab, Cat#: 75-156, 1:500 IC; 1:1,000 W), pan-Shank (NeuroMab, Cat#: 75-089, 1:1000 W), SVP-38 (Sigma, Cat#: S-5768, 1:400 IC), Rab9 (Millipore, Cat#: 552101, 1:500 IC), K48-specific ubiquitin (Millipore, Cat#: 05-1307, 1:1000 W), mouse-vGLUT1 (Millipore, Cat#: MAB5502, 1:1000 W), guinea pig vGLUT (Synaptic Systems, Cat#: AB5905, 1:1000 IC and IHC), vGAT (Synaptic Systems, Cat#: 131,011, 1:400 IC and IHC; 1:1000 W), Wnt5a (Abcam, Cat#: ab72583, 1:250 W), and Wnt7a (R&D Systems, Cat#: AF3008, 1:250 W). Soluble forms of Wnt5a, Wnt7a, sFRP-1, and sFz5c-Fc were obtained from R&D Systems and used at the final concentrations of 100 ng ml$^{-1}$ for Wnts and 1 µg ml$^{-1}$ for sFRP-1 and sFz5c-Fc. The final concentrations of various inhibitors used were 100 µM APV, 50 µM CNQX, 40 µM Bic, 10 µM MG132, 1 µM lactacystin, 5 µM z-DEVD-FMK, 10 µM TDZD-8, and 10 mM LiCl.

**Plasmid constructs.** HA-PRR7 was generated by inserting the HA epitope at a position 15 amino acid upstream of the C terminus of human PRR7 complementary (cDNA) to avoid the disruption of its interaction with PDZ-domain-containing proteins. HA-N-PRR7 was generated by putting the HA-tag at the N terminus of PRR7. PRR7-GFP was generated by putting EGFP at the C-terminal end of PRR7. PRR7 RNAi constructs were prepared by putting shRNA oligonucleotides into pSUPER.neo + gfp vector (OligoEngine). As a control RNAi, pSUPER.neo + gfp expressing shRNAs targeting luciferase was used.[43] The GSK3β knockdown construct was generated in pSUPER.neo + gfp by inserting GSK3β shRNA oligonucleotides as described previously. To generate PRR7 mutants (dC4 and C3S), shRNA-resistant PRR7 (PRR7-r), Rab11a DN, and Rab27b DN, site-directed mutagenesis was performed using the Quick Change System protocol (Agilent Technologies). pcDNA-Wnt5a-V5 and pcDNA-Wnt7a-V5 were gifts from Dr. Marian Waterman (Addgene plasmids #35930 and #35933) and myc-GPR177[34] was kindly provided by Dr. Wei Hsu.

**Transfection.** Dissociated hippocampal cultured neurons were transfected at 14–15 days in vitro (DIV) with Lipofectamine 2000 reagent (2 µg plasmids per well). For assessing overexpression effect, neurons were used <1 day post transfection (typically 18 h). For knockdown effect, neurons were used at 3 days post transfection. Biolistic transfection was used to transfect CA1 cells in the hippocampal slice culture, using Helios Gene Gun (Bio-Rad) and gold particles coated with plasmids including the transfection marker pEGFP-C1 (Clontech). HEK293 cells were transfected using Lipofectamine and OPTI-MEM with 1–2 µg plasmids per well.

**Immunocytochemistry.** Neurons were fixed by 5 min incubation in 2% formaldehyde/4% sucrose/1× phosphate-buffered saline (PBS), followed by 10 min incubation in cold methanol (−20 °C). Fixed neurons were incubated with primary antibodies diluted in 1× GDB (0.1% gelatin, 0.3% Triton X-100, 0.45 M NaCl, 17.7 mM sodium phosphate buffer, pH 7.4) overnight at 4 °C in a humidified chamber. After washing in 1× PBS three times for 10 min, bound antibodies were visualized by 1 h incubation with secondary antibodies at room temperature (RT). For surface staining, cells were fixed in 4% formaldehyde/4% sucrose/1× PBS for 5 min and incubated with antibodies diluted in ADB (4% horse serum, 0.1% bovine serum albumin (BSA), 1× PBS). For total staining, the formaldehyde-fixed cells were permeabilized with 0.2% Triton X-100/1× PBS for 10 min before incubation with antibodies.

**Image acquisition and analyses.** Images were captured by using a Nikon C1 plus laser scanning confocal microscope and ×60 objective (NA1.4). Acquired images (z-series stacks) were first converted to projection images (with maximal projection

option) for analyses. Both image acquisition and analyses were done in a blind manner. To measure puncta numbers and intensities per given neurons, 4–5 dendritic segments (~15–30 μm in length each) were selected from Txf and neighboring Nxf neurons, respectively, and their average values were used. After applying threshold, only puncta with more than three pixel sizes were counted and their pixel area and total and average intensity were also measured using SynPAnal software[52]. All data collected were transferred to Microsoft Excel for computation and graphical representation.

**Exosome preparation, sucrose gradient, and EM analyses.** CS of hippocampal neurons were collected and used for exosome preparation by differential centrifugation (cfg) performed at 4 °C[25]. Briefly, CS (conditioned medium) was subject to low-speed cfg ($2k \times g$ for 10 min), followed by a med-speed cfg ($20k \times g$ for 15 min). Supernatant from the med-speed cfg was filtered through surfactant-free cellulose acetate membrane syringe filters (0.2 μm pore size; ThermoFisher Scientific) and further subject to high-speed cfg at $100k \times g$ for 60 min. The resulting pellet was washed once in 1× PBS and re-centrifuged at $100k \times g$ for 60 min. Final pellets (P100) containing exosomes were resuspended in 1× PBS and saved at −80 °C. For sucrose gradient analyses, exosomes were first purified from CS harvested from four dishes (100 mm) plated with cortical neurons (4 million cells total; 40 ml). The P100 was resuspended in 1 ml of 0.211 M sucrose/3 mM imidazole (pH 7.4) buffer, loaded onto a sucrose gradient (0.211–2.255 M sucrose in 3 mM imidazole, pH 7.4), and subject to cfg at $100k \times g$ for 18 h using a swinging bucket rotor (Beckman SW32.1). After the cfg, 1 ml fractions were collected from the bottom of the tube, diluted with 2 ml of 3 mM imidazole, pH 7.4, and centrifuged at $100k \times g$ for 1 h to obtain pellets for subsequent western blot analyses. For EM analysis, the $100k \times g$ exosome pellets from fractions #8 were resuspended and fixed in 2% paraformaldehyde/0.1 M sodium phosphate buffer and mounted on Formvar-carbon-coated EM grid (Ted Pella, Cat#10700-F). Grid-mount exosomes were further fixed in 1% glutaraldehyde for 5 min. Contrasting and embedding of the samples were done using uranyl-oxalate and methylcellulose-uranyl acetate[25]. Immunogold labeling of exosomes were performed as described[25] using mouse anti-HA monoclonal antibody (12CA5, IgG$_{2bk}$, 10 μg ml$^{-1}$) and gold-conjugated Protein-A (10 nm, Ted Pella, Cat#15821; 1:100 dilution) diluted in the blocking buffer (1% (w v$^{-1}$) cold-water fish gelatin, 0.05% saponin, 1× PBS). Images were acquired using Hitachi 600 scanning electron microscope.

**CS and exosome treatment experiments.** Neurons were transfected at 14 DIV with various plasmids and CS was harvested after 24 h transfection. Conditioned medium of naive neurons (15 DIV) was replaced by the CS and neurons were further incubated for 0.5–24 h, fixed, and stained for PSD-95. For exosome treatment experiments, exosomes were obtained by collecting the P100 from the CS of HA-PRR7-transfected hippocampal neurons. Naive neurons (15 DIV) were treated with purified exosomes for the indicated time points and processed for immunocytochemistry. For quantification of exosomes, we used the amount of exosomes purified from the same number of neurons and also utilized Flotillin-1 or Alix as additional controls (by western blots). Typically, we used exosomes purified from the hippocampal neuron CS from 6 wells of 12-well plates, which contains 75,000 cells per well (total $4.5 \times 10^5$ cells). For exosome uptake experiments (Fig. 5h, i), we used exosomes purified from the CS of two 12-well plates (total $1.8 \times 10^6$ cells) and applied to neurons grown in 24-well plate (10,000 cells per 12 mm diameter coverslip). After 2 h incubation with exosomes, coverslips were extensively washed by gentle swing 15 times in ice-cold 1× PBS (200 ml), followed by fixing and immunocytochemistry.

**TCL preparation and western blotting.** Neurons were washed briefly in ice-cold 1× PBS after removing CS. TCL was prepared by adding 2× SDS sample buffer preheated at 65 °C directly to the wells. After boiling for 5 min, samples were loaded onto sodium dodecyl sulfate–polyacrylamide gel electrophoresis gels, transferred onto PVDF membranes (Immobilon-P, Millipore), and processed for immunoblotting after blocking in 6% nonfat-dried milk/1× Tris-buffered saline Tween-20. Blots were incubated with primary antibodies diluted in the blocking buffer overnight at 4 °C, followed by 1 h incubation with horse radish peroxidase-conjugated secondary antibodies (GE Healthcare) at RT. Bound antibodies were detected using ECL detection substrates (GE Healthcare or Biotool) and luminescent image analyzer (ImageQuant LAS4000, GE Healthcare). Uncropped scans of important blots are shown in Supplementary Figure 10.

**Immunoprecipitation.** For immunoprecipitation of PSD-95, neurons (from 6 wells) were lysed by using the Lysis buffer (50 mM Tris·Cl, pH 7.4, 75 mM NaCl, 2.5 mM EDTA, 2.5 mM EDTA, 2 mM dithiothreitol, 0.2% SDS, and protease and phosphatase inhibitor cocktail (ThermoFisher Scientific)) and boiled for 3 min. After cfg at $15k \times g$ for 10 min, supernatant was mixed with equal volume of 2× radioimmunoprecipitation assay (RIPA) buffer (20 mM Tris·Cl, pH 7.4, 2 mM EDTA, 300 mM NaCl, 2% Triton X-100, 1% sodium deoxycholate, and 1 mg ml$^{-1}$ BSA). Aliquots were incubated overnight at 4 °C with anti-mouse-PSD-95 antibodies (10 μg) pre-coupled to Protein-A-agarose beads, washed four times with 1× RIPA buffer (10 min each), and eluted in 2× SDS sample buffer by boiling.

**Electrophysiology.** Hippocampal neurons, plated at the density of $1.5 \times 10^5$ cells per coverslip, were transfected at 14 DIV with plasmids expressing various HA-PRR7 forms. After 18 h post transfection, transfected pyramidal neurons were identified by GFP fluorescence and morphological inspection. For mEPSCs, whole-cell patch recordings were performed by voltage-clamping neurons at −70 mV in bathing solution (in mM, 119 NaCl, 5 KCl, 2 CaCl$_2$, 2 MgCl$_2$, 30 glucose, 10 HEPES, pH 7.4, 300 mOsm) containing TTX (1 μM) and Bic (20 μM), continuously perfused at the rate of ~0.5 ml min$^{-1}$. Internal solution was composed of 140 K-gluconate, 5 KCl, 2 MgCl$_2$, 4 Mg-ATP, 0.3 Na$_2$-GTP, 0.2 EGTA, 10 HEPES, and adjusted to pH 7.2 and 290 mOsm. For mIPSCs, neurons were voltage clamped at −60 mV with internal solution (in mM, 153.3 CsCl, 1 MgCl$_2$, 4 Mg-ATP, 5 EGTA, 10 HEPES, pH 7.2) in bathing solution (146 NaCl, 2.5 KCl, 2 CaCl$_2$, 3 MgCl$_2$, 10 glucose, 10 HEPES, pH 7.4) containing TTX (2 μM), APV (50 μM), and NBQX (10 μM). mEPSCs and mIPSCs were acquired through a MultiClamp 700B amplifier (Molecular Devices), filtered at 2 kHz and digitized at 10 kHz. mEPSCs were detected and analyzed with the MiniAnalyses software (Synaptosoft). The following criteria were used to detect miniature PSCs: setting amplitude threshold to 5 pA, 10–90% rise time <3 ms (for mEPSCs) or rise time of 0.1–5 ms with half-width >2 ms (for mIPSCs).

For simultaneous dual whole-cell recordings, a single slice was removed from the culture insert 15–24 h post transfection and placed in the recording chamber and continuously perfused with artificial cerebrospinal fluid (in mM; 119 NaCl, 2.5 KCl, 1 NaH$_2$PO$_4$, 11 glucose, 26 NaHCO$_3$, 4 MgCl$_2$, 4 CaCl$_2$, 290 mOsm) containing 50 μM picrotoxin and 2 μM chloroadenosine, bubbled with a mixture of 5% CO$_2$, and 95% O$_2$. Glass pipettes (3–5 MΩ) were filled with an internal solution containing (in mM): 140 K-gluconate, 5 KCl, 2 MgCl$_2$, 10 HEPES, 0.2 EGTA, 4 Mg-ATP, 0.3 Na$_2$-GTP, 10 Na$_2$-phosphocreatine (pH 7.2 with KOH, 290 mOsm). Recordings were made from the transfected and non-transfected neighboring neurons under fluorescent and infrared-differential interference contrast optics. Stimulating electrodes (2 conductor platinum/iridium cluster microelectrode, 25 μm diameter; FHC) were placed about 200 μm on either side of the recorded cells. The evoked EPSCs were measured at −60 mV with a Multiclamp 700B amplifier (Axon Instruments). Data acquisition and analysis were performed using DigiData 1440A digitizer and analysis software pClamp10 (Axon Instruments). Signals were filtered at 2 kHz and sampled at 10 kHz. Stimulation pulses were provided at 0.1 Hz. An average of 10–20 consecutive traces was collected and averaged for EPSCs.

**Statistical analysis.** All values represent means ± SEM, unless otherwise indicated. All transfection experiments were done in duplicate or triplicate using independent neuron cultures. For multiple group comparisons, one-way or two-way analysis of variance with Tukey's multiple comparison post hoc test were used using the GraphPad Prism software. Student's t test (unpaired, two-tailed, assuming unequal variance) was used to determine the statistical significance for the pair. $P < 0.05$ was considered significant.

**Data availability.** The data that support the findings of this study are available from the corresponding author upon reasonable request.

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

## Acknowledgements

We thank Clive Wells at the MCW Electron Microscopy Core Facility for EM image acquisition and Dr. Wei Hsu (University of Rochester Medical Center) for providing myc-GPR177 construct. We also appreciate Casey Vickstrom and Christopher Olsen for reading the manuscript. This research was supported by Charles Jacobus Family Foundation, funding provided through the Research and Education Program, a component of the Advancing a Healthier Wisconsin endowment at the Medical College of Wisconsin (S.H.L.), and NIH/DA035217 and MH101146 (Q.-s.L.).

## Author contributions

S.H.L. and C.-H.K. conceived the study; S.H.L. designed experiments; S.M.S., P.Z., H.-T. K., J.M.K., D.-I.K., and S.H.L. conducted experiments; S.M.S., D.-W.K., C.-Y.Y, Q.-s.L., and S.H.L. analyzed data; D.-W.K and W.D.H developed reagents; S.H.L. wrote the paper.
