## [Peer Review File · Nature Communications]

Editorial Note: An image has been redacted as indicated to protect copyright claims. The figure can be viewed in the original publication, listed in the place of redaction.

Reviewers' comments:

Reviewer #1 (Remarks to the Author):

The manuscript by Lee et al reports the novel role of PRR7, a single pass membrane protein, in the regulation of synapse number. The authors proposed that PRR7 is secreted through exosomes to suppress the secretion of Wnt proteins resulting in changes in synapse number. However, PRR7 might also regulate synapse number through a cell autonomous mechanism that does not require Wnt modulation.

In this new study, the authors present clear evidence that PRR7 is present in exosomes, vesicles that also contain Wnt7a and Wnt5a proteins, which are powerful synaptogenic factors. The data also demonstrates that PRR7 is a negative regulator of excitatory synapse number in hippocampal neurons. The authors proposed that PRR7 is released from neurons in exosomes affecting synapse number on cell autonomous and cell non-autonomous mechanisms. Vivian Budnik has previously demonstrated a role for exosomes in Wnt release at the *Drosophila* neuromuscular synapse. However, the function of exosomes at vertebrate synapses has not been documented. This aspect of the paper is clearly novel.

The current paper presents unexpected and potentially very interesting results. However, as I read the paper my enthusiasm waned due to the lack of proper controls and poor presentation of some of the data. In addition, the model proposed for the flipping of the PRR7 protein at the plasma membrane in cells where exosomes have delivered PRR7 is too premature. These important issues need to be addressed before this paper is further considered by Nature Communications.

Specific comments:

- 1) In Figure 1, the authors examined the impact of neuronal activity on the levels of PRR7 using APV. It would be important to test TTX, KCl or chemical LTP.
- 2) The findings, obtained from both overexpression and loss of function studies, are complex. Based on overexpression studies, the authors concluded that PRR7 has a non-cell autonomous phenotype (Fig. 2) but the images presented are not convincing. Better images should be presented.
- 3) Figure 2a, the neurons expressing HA-PRR7 seem less healthy than controls. The authors should express GFP in control as well as in HA-PRR7-expressing cells to make a better comparison.
- 4) The author claimed that the impact of PRR7 is non-cell autonomous, how do the authors explain that the KD of PRR7 on spare neurons has an effect? Neighbouring neurons expression normal levels of PRR7 could compensate for the loss of function of PRR7 if the authors are correct that PRR7 functions non-cell autonomously. Proper quantification of synapses on distal neurons from the transfected cell should be provided.

- 5) How do the authors evaluate dendritic spines in Fig 2c? According to the images presented, the authors compared control GFP expressing cells with cells expressing just HAPRR7 but without GFP. This experiment was not properly designed and therefore the conclusion that PRR7 affects spines is premature.
- 6) The authors should present supporting data on the impact of PRR7 loss of function or overexpression in the intact hippocampus (ex vivo or in vivo).
- 7) Figure 3, the traces do not fit with the values of the mEPSC frequency and amplitude presented in the graphs. Is the scale provided correct?
- 8) PRR7 overexpression has a moderate impact on inhibitory synapse number on cells that overexpressed this protein but not on neighbouring cells. If the model is correct that PRR7 is released in exosomes to affect neighbouring cells, how do the authors explain this result? Given the impact of PRR7 on Wnt5a, which affects inhibitory synapses, one would expect an effect on inhibitory synapses in neighboring cells.
- 9) The analyses of cell death in Supplementary Figure 3 were done in much younger neurons than those used for functional analyses in previous figures. The authors should properly demonstrate the impact of PRR7 on cell death in mature neurons.
- 10) The experiments in Figure 4 using the supernatant of cells (containing exosomes) are convincing. However, I do not understand how spot application of exosomes isolated from HA-PRR7 expressing cells do not have an impact on PSD95 puncta (Fig. 4h). The authors need to address this.
- 11) Figure 4i, the difference in the fluorescence levels between control and HA-PRR7 expressing cells is so extreme that a proper counter-staining should be included.
- 12) To claim that the exosomes containing PRR7 fuse with the plasma membrane (end of first paragraph, page 10) requires more demonstration such as surface staining after extensive washes or the internalization of PRR7 into endosomes on neighbouring cells alternatively the authors could considering in using Superfluorin (SEP).
- 13) Second paragraph page 10, the authors wrote that HA-N-PRR7 showed vesicular staining patterns of exosomes (Suppl. Figure 4c). This conclusion is premature at this point in the paper.
- 14) I fail to understand why after 3 hours of transfection of HA-PRR7 (Supplementary Figure 4e), no effect is observed on the number of synapses. This seems to contradict previous experiments in the paper.
- 15) I question the model presented in Supp. Figure 4 f. According to the model, fusion of PRR7-containing exosomes results in the flip of the PRR7 protein in the plasma membrane such that the intracellular domain (N terminus) is now extracellular and the previously extracellular domain is now intracellular. This type of change in the topology of a membrane protein would be an extraordinary event resulting in the complete change in the interactions of proteins on both sides of the plasma membrane. This model requires serious evaluation and demonstration.
- 16) Figure 5 b shows that expression of DN Rab11 in PRR7 expressing cells leads to only a partial recovery of PSD95 puncta. However, this data does not fit with the quantification presented in Fig, 5c.
- 17) If Rab27b is important for exosome formation, expression of DN Rab27b should also affect Wnt secretion, which are released through exosomes. Therefore, this manipulation should not affect synapse number.
- 18) Figure 7, the quantification presented does not match the images. For example, the

images presented do not show an effect by Wnt5a and Wnt7a on PSD95 puncta but the graph does.

Minor comments:

The authors should properly quote papers on the role of Wnts in synapse integrity/maintenance in the introduction and discussion (Marzo et al, Current Biology 2016) and also the finding that PRR7 regulates the trafficking of Frizzled receptors (Kim et al, Scientific Report 2017).

Reviewer #2 (Remarks to the Author):

The manuscript from Lee and colleagues describes very interesting findings regarding the control of excitatory synapse numbers by exosomal Wnt inhibitor PRR7. The authors use Hek293 cells, rat hippocampal neurons in culture, organotypic slices and KO mice tissue to show a non-autonomous effect of exosomal PRR7 on excitatory synapse numbers. While PRR7 was described in an EMBO paper from 2016 as a neuronal death promoting factor (Kravchick et al. 2016), its connection to exosomes and Wnt signaling has not been described before, which makes it a very novel study. Moreover, especially interesting are the non-autonomous effects observed in surrounding neurons induced by these PRR7 exosomes. This will impact different fields such as intercellular communication, Wnt signaling and regulation of synapse maintenance and the Extracellular Vesicles field. However, the structure of the manuscript is less appealing and lacks focus in the middle section. There is redundant content in Figure 2, 4 and 5, e.g. the results from coculture experiment versus supernatant containing exosomal PRR7 could be combined and the text rewritten for clarity. While on the other hand, the text elaborates a lot about data only shown in Supplementary Figure 2 and 3, which should be shown at least in parts in main figures, as well as the in vivo data from Supplementary Figure 7.

It is quite challenging still to quantify exosomes. It is done based on either: 1. same cell numbers secreting exosomes, 2. Protein content of exosome samples, 3. number of secreted and purified (by Nanoparticle tracking) or 4. Based on a set of exosomal markers in Western blots. From the manuscript, it is not clear, whether and how the authors quantified the amount of exosomes secreted from neurons. How much exosomes were added to cultures in uptake assays and functional assays. Although the quantification is based on PSD95 puncta or intensity/ μm , it is necessary to mention how exosomes were quantified and how many were added. This will greatly add to the reproducibility of Extracellular Vesicles research. In summary, I would reconsider the manuscript for publication in Nature Communications with appropriate changes made.

More specifically:

1. What is the explanation for the significant effect of HA-PRR7 on inhibitory synapses in transfected neurons in Supplementary Figure 2, while RNAi and GFP-PRR7 have no phenotype? Do inhibitory synapses require Wnt signals?
2. In the 2016 EMBO J paper from Kravchick et al. PRR7 inhibits the ubiquitination of c-Jun,

how can this be reconciled with the effect of overexpression of PRR7 on general poly-ubiquitination in Figure 6?

3. Additional experiments I would like to see in a revised manuscript:

The connection to Wnt signaling is comprehensible, but very focused on the experimental system and slightly preliminary.

Figure 7g Does PRR7 overexpression reduce exosomal Wnt5a and Wnt7a secretion? Could paracrine canonical Wnt reporter assays be used to show the reciprocal effects of PRR7 and Wnt secretion on canonical Wnt activity? What is the pathway involved? Is there a direct binding of PRR7 to Wnts? Are PRR7 and Wnts present on the same exosomes, which could be easily detected by HA-PRR7 precipitation of exosomes and probing for Wnts. Is there an influence of PRR7 on Evi/WLS or surface levels of Wnt? Some more mechanistic insight would clearly strengthen this part of the reasoning.

4. Some minor comments:

Line 93: What are APV, CNQX? Please specify in the text.

Line 183: in or on exosomes

Line 252: Do you mean Figure 6a (inset)?

Figure 6 Quantification of PRR7 overexpression effect on polyubiquitination in Fig 6c or mention in the text as in Fig 6e.

Line 402 PRR7

Line 573 Which company provided active Wnt5a and Wnt7a?

Reviewer #3 (Remarks to the Author):

The paper presented an interesting set of data on PRR7 regulating synapse development. PRR7 was shown to be in exosomes and inhibit synapse formation. Analysis of synapses by morphology and electrophysiology experiments appeared to be solid. PRR7 was shown to act by blocking Wnt secretion from exosomes (thus activating GSK3b and PSD degradation). The hypothesis appeared to be novel and may be useful to peers in the field. There are a few points that need to be tightened up prior to publication.

1. The title "Reciprocal Control of Excitatory Synapse Numbers by Wnt and Novel Wnt Inhibitor Secreted on Exosomes" is superficial and misleading. It does not even mention PRR7.

2. Wnt release from NMJ is not an accurate statement. The cited papers were dealing with *Drosophila* NMJ, not mammalian NMJ. These two synapses used completely different neurotransmitters.

3. Figure 1. Punctas or dots "outside" of neurons – could they be axons or dendrites of unlabeled neurons? Do they colocalize with PSD95 or other synaptic markers? How about exosome markers?

4. What is the evidence that the vesicle in EM image is the one containing PRR7? Such claim cannot be made unless immunoEM is done. EM images of vesicles where PRR7 is low should be provided to conclude that PRR7-containing vesicles are unique.

5. Biochemical fraction experiments need control markers. It would be important to show that some markers are not clusters in the middle fractions (where PRR7 is). At the moment, all proteins tested were clustered in the middle fractions.

6. Figure 2. Overexpressing PRR7 reduces synapses in non-transfected (Nxf) neurons – Is

this a toxic effect or effect of overexpressed PRR7 in transfected (Txf) neurons (both pyramidal and interneurons)? This should be addressed, considering others have some that PRR7 is pro-apoptotic. The authors have shown that neurons are not dead, but were they healthy? If not, synapses may disappear.

7. As shown in later figures, PRR7 overexpression in pyramidal neurons and in interneurons alter their activity. Thus, the effect on Nxf neurons could be a secondary effect? Can this effect be blocked by exosome inhibitors?

8. In various places, PRR7-enriched exosomes were referred. It would be important to include PRR7-poor exosomes as control.

9. Can PRR7's effect be tested directly by tossing purified, recombinant PRR7 or different domains of it on neurons? This is important as it will change the conclusion. PRR7 could be a marker of exosomes and may be necessary for exosomes' effect on synapses, but not PRR7 itself directly regulates synapses. Along the same line, do exosomes of HEK293 cells have similar effect to those of Txf neurons? Why not? Don't they express Wnts?

10. The linear pathways of PRR7-Wnt seemed to be over-simplified. The evidence that Wnts work directly downstream of PRR7 is weak although they could "rescue" the phenotypes. There could be many steps in between.

11. Wnt5a was shown to promote the formation of inhibitory synapses (Cuitino, 2010). PRR7's effect was specific for excitatory synapses, but not GABA synapses. However, Wnt5a was believed to be an effector of PRR7. Something does not connect here.

Responses to Reviewers' Comments

First of all, we would like to thank the reviewers for the encouraging and constructive comments on our manuscript. We hope that the revised manuscript alleviates the reviewers' previous concerns. Modified or newly added texts in the revised manuscript are marked by red font. Our point-by-point responses to the specific comments are below.

Reviewer #1 (Remarks to the Author):

The manuscript by Lee et al reports the novel role of PRR7, a single pass membrane protein, in the regulation of synapse number. The authors proposed that PRR7 is secreted through exosomes to suppress the secretion of Wnt proteins resulting in changes in synapse number. However, PRR7 might also regulate synapse number through a cell autonomous mechanism that does not require Wnt modulation. The current paper presents unexpected and potentially very interesting results. However, as I read the paper my enthusiasm waned due to the lack of proper controls and poor presentation of some of the data. In addition, the model proposed for the flipping of the PRR7 protein at the plasma membrane in cells where exosomes have delivered PRR7 is too premature. These important issues need to be addressed before this paper is further considered by Nature Communications.

We have tried to provide proper controls and better explanation of the data. For issues regarding the topology in model, please see our responses to the specific comments below (#15).

Specific comments:

1) In Figure 1, the authors examined the impact of neuronal activity on the levels of PRR7 using APV. It would be important to test TTX, KCl or chemical LTP.

– We now provide new data showing that TTX greatly reduces the levels of PRR7 in exosomes ($35 \pm 7\%$ of control; Supplementary Fig. 1d), which further supports our notion that PRR7 is released on exosomes in an activity-dependent manner.

2) The findings, obtained from both overexpression and loss of function studies, are complex. Based on overexpression studies, the authors concluded that PRR7 has a non-cell autonomous phenotype (Fig. 2) but the images presented are not convincing. Better images should be presented.

– We substantially revised Figure 2 to better illustrate the non-cell autonomous effect of PRR7 in non-transfected (Nxf) neighboring neurons by providing representative images of Nxf. In addition, we added separate grey-scale images of red channels used for PSD-95 or vGLUT puncta in Nxf images.

3) Figure 2a, the neurons expressing HA-PRR7 seem less healthy than controls. The authors should express GFP in control as well as in HA-PRR7-expressing cells to make a better comparison.

– The “less-healthy” impression of HA-PRR7-expressing neurons is probably due to the unusual staining patterns of PRR7 (speckles mainly distributed in the soma and proximal major dendrites). We do not think they are sick also because HA-PRR7 overexpression did not increase cleaved caspase-3 levels in Txf and Nxf neurons (Supplementary Fig. 3). To convince the reviewer, we now added additional images of HA-PRR7-expressing hippocampal neurons stained for MAP2 (Supplementary Fig. 1a) or cell-fill

marker β -Gal (Supplementary Fig. 3a), which show overall normal neuronal morphology (dendritic arborization and tapering) of PRR7-transfected neurons. In fact, one-replicate set of experiments quantified in Fig. 2b,c (done in triplicates) was obtained from the co-transfection experiments (β -Gal was used as a fill-marker instead of GFP to distinguish fluorescence from transfected PRR7-GFP) as the reviewer suggested. We chose to show the images in Fig. 2a,c without the cell-fill marker for simplicity and space-saving purpose. We also would like to point out that representative images of PSD-95 in PRR7-expressing neurons co-stained with cell fill-markers are shown in Fig. 2g (the middle panel) and Fig. 6b (2nd panel, CFP + HA-PRR7).

4) The author claimed that the impact of PRR7 is non-cell autonomous, how do the authors explain that the KD of PRR7 on spare neurons has an effect? Neighboring neurons expression normal levels of PRR7 could compensate for the loss of function of PRR7 if the authors are correct that PRR7 functions non-cell autonomously. Proper quantification of synapses on distal neurons from the transfected cell should be provided.

– We posit that neurons secrete both Wnts and PRR7 and the shifts in their balance determine synaptic growth or weakening. We suggested that PRR7 loss (by sparse knockdown) leads to increased release of Wnts (Fig. 8g; ~2 times higher than control levels) from the neurons, which overpowers endogenous secreted PRR7. Regarding the effect of PRR7 on distal neurons, we now provide representative low-magnification images of neurons (acquired using 10 \times objective; Supplementary Fig. 2a) showing significant reduction of PSD-95 staining intensity in the whole field ($43.5 \pm 5.6\%$ of control, $n = 6$, unpaired t-test, $P = 0.018$). However, we were unable to accurately quantify PSD-95 puncta densities due to low resolving power of the images. Therefore, we instead analyzed the effects of PRR7 on synapses in neighboring non-transfected neurons (Nxf) in higher magnification images (60 \times objective), which were quantified in Fig. 2 (b,d,f, and h).

5) How do the authors evaluate dendritic spines in Fig 2c? According to the images presented, the authors compared control GFP expressing cells with cells expressing just HAPRR7 but without GFP. This experiment was not properly designed and therefore the conclusion that PRR7 affects spines is premature.

– As the reviewer pointed out, the effects of HA-PRR7 on dendritic spines were indeed analyzed from images of neurons co-transfected with β -Gal. To clearly indicate the fact, we also provide the representative images of dendritic spines in Fig. 2i.

6) The authors should present supporting data on the impact of PRR7 loss of function or overexpression in the intact hippocampus (ex vivo or in vivo).

– We would like to point out that, in the original submission, the effects of PRR7 overexpression and knockdown in intact hippocampal slices (*ex vivo*) were shown in Fig. 3d,e. Moreover, we also showed the effect of PRR7 loss of function *in vivo* using PRR7 heterozygote KO mice in Supplementary Fig. 7 (now Supplementary Fig. 9). In the revised manuscript, we added a new data set showing the significant increase of excitatory synapse numbers in the hippocampus (stratum radiatum of CA1 area) of homozygous *Prr7* knockout mice, which is measured by the numbers of vGLUT1 puncta (Supplementary Fig. 9c) using immunohistochemistry of intact brain slices.

7) Figure 3, the traces do not fit with the values of the mEPSC frequency and amplitude presented in the graphs. Is the scale provided correct?

– We carefully re-examined the traces and confirmed that the scale is correct and the rep traces match the average data points.

8) PRR7 overexpression has a moderate impact on inhibitory synapse number on cells that overexpressed this protein but not on neighbouring cells. If the model is correct that PRR7 is released in exosomes to affect neighbouring cells, how do the authors explain this result? Given the impact of PRR7 on Wnt5a, which affects inhibitory synapses, one would expect an effect on inhibitory synapses in neighboring cells.

– This is an important question. Cuitino et al. (2010) is the only published paper describing the effect of Wnt5a on GABAergic synapses in hippocampal neurons. Specifically, they reported that short term (up to 1 h) application of exogenous Wnt5a (at undefined concentration) increased surface expression of GABA_A receptors and mIPSC amplitudes but did NOT affect mIPSC frequencies. They also showed that the non-canonical Wnt5a–CaMKII pathway is likely involved in the process. It was suggested that Wnt5a promotes the recycling of GABA_A receptors to preexisting inhibitory synapses. Importantly, in their experiments, the blockade of Wnt signaling by sFRP did not affect GABAergic synapses significantly. These results are consistent with the findings that blockade of the canonical Wnt signaling by inducible Dkk1 expression did not affect hippocampal GABAergic synapse in vivo (Marzo et al., 2016). Therefore, exogenous Wnt5a did not change the number of GABAergic synapses and the role of endogenous Wnt5a in the modulation of GABAergic synapse numbers still remains to be demonstrated.

Our model suggests that PRR7 overexpression reduces Wnt5a secretion. However, Wnt5a reduction is not expected to affect GABAergic synapses significantly, as demonstrated by Cuitino et al. (2010) and Marzo et al. (2016). The decrease of GABAergic synapse numbers in PRR7-transfected neurons (cell-autonomous effect), as we described in Discussion, could be explained by either inhibition of the non-canonical Wnt–CaMKII pathway or homeostatic plasticity induced by the severe reduction of excitatory activity (please note that PRR7 transfected neurons lose virtually all excitatory synapses but Nxf neurons still have reduced but significant numbers of excitatory synapses) and/or by the strong activation of GSK3β. On the other hand, PRR7 knockdown would increase Wnt5a secretion. However, the sensitivity (EC₅₀) and long-term (24 h) effect of Wnt5a on GABAergic synapses is unknown and could be also influenced by a mechanism such as homeostatic control. Therefore, our model is not at odds with the findings of Cutini et al. At present, there is much unknown for the detailed molecular mechanisms of PRR7 and Wnt5a action on the GABAergic synapses. Further intensive studies are required to fully understand the physiological role of Wnt5a and PRR7 in the modulation of GABAergic synapses.

9) The analyses of cell death in Supplementary Figure 3 were done in much younger neurons than those used for functional analyses in previous figures. The authors should properly demonstrate the impact of PRR7 on cell death in mature neurons.

– We used the same age mature neurons (div 14-16) for the experiments, like all other functional experiments. They appear smaller (and thus perhaps look younger) because images represent a wider field of view than images in other data to show more number of cells (please note the differences in scale bars).

10) The experiments in Figure 4 using the supernatant of cells (containing exosomes) are convincing. However, I do not understand how spot application of exosomes isolated from HA-PRR7 expressing cells do not have an impact on PS95 puncta (Fig. 4h). The authors need to address this.

– We provide higher magnification of the original images to show better that the application of PRR7 exosomes reduced intensity and density of PSD-95 puncta in Fig. 5h.

11) Figure 4i, the difference in the fluorescence levels between control and HA-PRR7 expressing cells is so extreme that a proper counter-staining should be included.

– The images of Fig. 4i were intensity profile images that were created by artificially converting the intensity of each pixel to a gradient of pseudo-colors. Therefore, control cells have very low intensities (background levels, close to black color) compared to cells treated with HA-PRR7-exosomes. We now replaced the images with new ones counter-stained with MAP2 that show diffuse HA immunofluorescence in the dendrites (Fig. 5i; please see also our response to #12 point below).

12) To claim that the exosomes containing PRR7 fuse with the plasma membrane (end of first paragraph, page 10) requires more demonstration such as surface staining after extensive washes or the internalization of PRR7 into endosomes on neighbouring cells alternatively the authors could consider in using Superfluorin (SEP).

– The images in Fig. 4h,i and Western blotting data (Fig. 4j) in the original submission were indeed obtained after extensive washing with PBS. We apologize for not providing the detailed methods of the experiments. To show more convincing data on the membrane fusion of exosomal PRR7, we now provide new images showing both total (with membrane permeabilization) and surface anti-HA immunofluorescent staining of hippocampal neurons that were incubated with exosomes purified from the culture supernatant of HA-PRR7-transfected neurons (Fig. 5i). These images show diffuse HA staining in the dendrites (determined by co-staining with MAP2) only in total but not in surface staining, suggesting the fusion of HA-PRR7 containing exosomes with the plasma membrane of recipient neurons. Similar diffuse dendritic staining of HA-PRR7 was also observed in non-transfected neighboring neurons (Supplementary Fig. 1a; marked by yellow dotted lines). These data are consistent with the membrane topology model of HA-PRR7 after plasma membrane fusion (Supplementary Fig. 5f), in which HA epitope is facing intracellular space of recipient neurons.

13) Second paragraph page 10, the authors wrote that HA-N-PRR7 showed vesicular staining patterns of exosomes (Suppl. Figure 4c). This conclusion is premature at this point in the paper.

– We modified the text to “fine vesicular staining patterns at the extracellular area of transfected neurons.”

14) I fail to understand why after 3 hours of transfection of HA-PRR7 (Supplementary Figure 4e), no effect is observed on the number of synapses. This seems to contradict previous experiments in the paper.

– The faster time course of PRR7 effect on PSD-95 in Figure 5 is observed with either replacement of culture supernatant or direct application of exosomes with high levels of PRR7. On the other hand, for the transfection experiment shown in Supplementary Fig. 4e, it requires a significant amount of time (for transgene mRNA expression and translation) to obtain overexpression of the PRR7 transgene, which would be naturally variable among transfected neurons (depending on how much transgene DNA they absorbed). In fact, there was noticeable reduction in the PSD-95 puncta in some other transfected neurons of the same batch experiment (data not shown). Since the point of Supplementary Figure 4e is to show that PRR7 does not traffic to synapses before it gets secreted to extracellular space, we picked up the one with the least effect on PSD-95 to clearly mark excitatory synapses. Therefore, we do not think the presence of PSD-95 puncta in PRR7-transfected neurons in the figure contradicts other experiments.

15) I question the model presented in Supp. Figure 4 f. According to the model, fusion of PRR7-containing exosomes results in the flip of the PRR7 protein in the plasma membrane such that the intracellular domain (N terminus) is now extracellular and the previously extracellular domain is now intracellular. This type of change in the topology of a membrane protein would be an extraordinary event resulting in the complete change in the interactions of proteins on both sides of the plasma membrane. This model requires serious evaluation and demonstration.

– As illustrated in the figure shown in the right, when vesicle fuses with plasma membrane, the topology of transmembrane proteins is conserved with respect to extracellular and intravesicular/intracellular sides. In the proposed model, the membrane topology of PRR7 is also conserved.

[Redacted: Fig. 2 from Martens, S. and McMahon, H.T. Mechanisms of membrane fusion: disparate players and common principles. *Nat. Rev. Mol. Cell Biol.* 2008; 9: 543–556.]

However, apparent changes in the topology happens when PRR7

containing endocytic vesicles become intraluminal vesicles of MBVs (via reverse topology membrane scission; Schonenberg et al., *Nat. Rev. Mol. Cell Biol.* 18, 2017): cytoplasmic side of the endocytic vesicles becomes extracellular side after the fusion of MVBs to plasma membrane. Therefore, we did not arbitrarily change the membrane topology of PRR7. Although we agree with the reviewer that the model requires further experiments to fully validate (outside the scope of current manuscript), we think that the model represents the best composition of the available knowledge so far and would be useful for readers.

16) Figure 5 b shows that expression of DN Rab11 in PRR7 expressing cells leads to only a partial recovery of PSD95 puncta. However, this data does not fit with the quantification presented in Fig, 5c.

– We modified the figures (Now Fig. 6b) to show additional grey-scale images of PSD-95 puncta in Nxf neurons. Now the images in the figure match better to the quantified data, showing no significant recovery of PSD-95 in Rab11b DN Txf and Nxf neurons. We now added a line in the text: these data suggest that “Rab27b, not Rab11, is involved in the exosomal secretion of PRR7 by neurons”. These findings are also consistent with the literature showing that different species of Rab protein are required for exosome biogenesis and secretion in various cell types.

17) If Rab27b is important for exosome formation, expression of DN Rab27b should also affect Wnt secretion, which are released through exosomes. Therefore, this manipulation should not affect synapse number. – This is the exactly what the data show. Co-overexpression of Rab27b DN completely prevented the effect of PRR7 overexpression on the synapse in neighboring Nxf neurons (Fig. 6b-d). However, the Rab27DN did not block the loss of PSD-95 puncta in transfected neurons, suggesting that cell autonomous effect of PRR7 is not solely dependent on its exosomal secretion and might be mediated by unidentified additional mechanisms as we discussed in the text (Discussion).

18) Figure 7, the quantification presented does not match the images. For example, the images presented do not show an effect by Wnt5a and Wnt7a on PSD95 puncta but the graph does.

– We carefully re-examined the images and found that they do match to the quantified data. We can only

guess that the impression of mismatch was originated from the small size images used for the PSD-95 puncta in the Figure, which may not show smaller PSD-95 puncta very well.

Minor comments:

The authors should properly quote papers on the role of Wnts in synapse integrity/maintenance in the introduction and discussion (Marzo et al, Current Biology 2016) and also the finding that PRR7 regulates the trafficking of Frizzled receptors (Kim et al, Scientific Report 2017).

– Both papers are now cited and we also added a line discussing the findings from zebrafish PRR7.

Reviewer #2 (Remarks to the Author):

The manuscript from Lee and colleagues describes very interesting findings regarding the control of excitatory synapse numbers by exosomal Wnt inhibitor PRR7. The authors use Hek293 cells, rat hippocampal neurons in culture, organotypic slices and KO mice tissue to show a non-autonomous effect of exosomal PRR7 on excitatory synapse numbers. While PRR7 was described in an EMBO paper from 2016 as a neuronal death promoting factor (Kravchick et al. 2016), its connection to exosomes and Wnt signaling has not been described before, which makes it a very novel study. Moreover, especially interesting are the non-autonomous effects observed in surrounding neurons induced by these PRR7 exosomes. This will impact different fields such as intercellular communication, Wnt signaling and regulation of synapse maintenance and the Extracellular Vesicles field.

However, the structure of the manuscript is less appealing and lacks focus in the middle section. There is redundant content in Figure 2, 4 and 5, e.g. the results from coculture experiment versus supernatant containing exosomal PRR7 could be combined and the text rewritten for clarity.

– These Figures are used to convey distinct points using different approaches and thus we respectfully disagree with the reviewer's opinion. Figure 2 shows the effect of PRR7 overexpression or knockdown on the excitatory synapses in cultured hippocampal neurons using immunocytochemistry. The content of Fig. 4 (Fig. 5 in revision) illustrate that exosomes containing high levels of PRR7 have the ability to eliminate excitatory synapses in neurons. On the other hand, Fig. 5 (Fig. 6 in revision) shows that exosomal secretion of PRR7 is required for the non-cell autonomous effect of PRR7. Therefore, the combined contents of Fig.4 and 5 establish the sufficiency and necessity of exosomal PRR7 in the regulation of excitatory synapses in neighboring neurons. We also modified the text to clarify these points.

While on the other hand, the text elaborates a lot about data only shown in Supplementary Figure 2 and 3, which should be shown at least in parts in main figures, as well as the in vivo data from Supplementary Figure 7.

– Following the suggestion, we moved Supplementary Figure 2 in the original submission to main Figures (now Figure 3). However, we kept Supplementary Figures 3 and 7 as Supplementary Figures because they are either essentially negative control experiments assessing cellular death and supplement the main findings from the in vitro experiments, respectively.

It is quite challenging still to quantify exosomes. It is done based on either: 1. same cell numbers secreting exosomes, 2. Protein content of exosome samples, 3.number of secreted and purified (by

Nanoparticle tracking) or 4. Based on a set of exosomal markers in Western blots. From the manuscript, it is not clear, whether and how the authors quantified the amount of exosomes secreted from neurons. How much exosomes were added to cultures in uptake assays and functional assays. Although the quantification is based on PSD95 puncta or intensity/ μm , it is necessary to mention how exosomes were quantified and how many were added. This will greatly add to the reproducibility of Extracellular Vesicles research.

– As the reviewer pointed out, we also had difficulties in quantifying exosomes using protein assays, which is due to the small amounts of purified exosomes from primary hippocampal neuron cultures. Therefore, we used the same amount of exosomes purified from **the same number of neurons** and utilized exosome marker Flotillin-1 or Alix as equal loading controls (by western blots). For sucrose gradient analyses, we used exosomes purified from culture supernatants harvested from four 100 mm dishes plated with cortical neurons (4 million cells). In exosome treatment experiments, we used exosomes purified from the hippocampal neuron culture supernatant from 6 wells of 12-well plates (total of 4.5×10^5 cells), except the exosome uptake experiments (in Fig 5h,I; from 1.8×10^6 cells). We have added the information in the Method section.

In summary, I would reconsider the manuscript for publication in Nature Communications with appropriate changes made.

More specifically:

1. What is the explanation for the significant effect of HA-PRR7 on inhibitory synapses in transfected neurons in Supplementary Figure 2, while RNAi and GFP-PRR7 have no phenotype? Do inhibitory synapses require Wnt signals?

– It is yet unclear whether inhibitory synapses require Wnt5a signals for its biogenesis and maintenance. Please refer to our responses to Reviewer 1's Specific Comments #8 above for detailed explanations.

2. In the 2016 EMBO J paper from Kravchick et al. PRR7 inhibits the ubiquitination of c-Jun, how can this be reconciled with the effect of overexpression of PRR7 on general poly-ubiquitination in Figure 6?

– According to the EMBO J paper, PRR7 inhibits C-Jun ubiquitination but the underlying molecular mechanism is unknown. In general, protein ubiquitination occurs via a mechanism SPECIFIC to substrate proteins: for example, their phosphorylation. PRR7 could inhibit the signaling pathway and/or promote the poly-ubiquitination of protein components that are involved in C-Jun ubiquitination. Therefore, it is not surprising that PRR7 promotes the ubiquitination of other proteins. Moreover, we did not intend to propose that PRR7 increases poly-ubiquitination of ALL neuronal proteins. In fact, PRR7-overexpression did not reduce the total protein level of β -catenin that is known to be poly-ubiquitinated and degraded by proteasomes (Fig. 8e,f). Therefore, to avoid confusion, we modified the text to specify that PRR7 promotes poly-ubiquitination (degradation) of PSD (synaptic) **scaffolding** proteins.

3. Additional experiments I would like to see in a revised manuscript:

The connection to Wnt signaling is comprehensible, but very focused on the experimental system and slightly preliminary. Figure 7g Does PRR7 overexpression reduce exosomal Wnt5a and Wnt7a secretion. Could paracrine canonical Wnt reporter assays be used to show the reciprocal effects of PRR7 and Wnt secretion on canonical Wnt activity? What is the pathway involved? Is there a direct binding of PRR7 to

Wnts? Are PRR7 and Wnts present on the same exosomes, which could be easily detected by HA-PRR7 precipitation of exosomes and probing for Wnts. Is there an influence of PRR7 on Evi/WLS or surface levels of Wnt? Some more mechanistic insight would clearly strengthen this part of the reasoning.

– These are very interesting and important questions, which would require an extensive research to understand completely. Nonetheless, we have performed additional experiments to provide some mechanistic insight. As the reviewer commented, we now show that PRR7 overexpression greatly reduces Wnt7a secretion (Supplementary Fig. 7a). Unfortunately, we were unable to determine whether PRR7 and Wnts are present on the same exosomes because PRR7 inhibits exosomal secretion of Wnts. The suggested immunoisolation of PRR7-containing exosomes was also not feasible since the epitopes of PRR7 antibodies (both endogenous and HA-PRR7) are not surface-exposed in exosomes (face intravesicular space). However, we examined the effect of PRR7 on the surface expression of Wnts and the localization of GPR177 (mouse ortholog of Evi/Wls). We found that, when expressed in HEK293 cells, PRR7 strongly blocks the surface expression of Wnt7a (Supplementary Fig. 7b). In addition, PRR7 dramatically inhibited the localization of GPR177 in the dendritic spines of neurons and promoted confinement to vesicular structures in the soma which colocalized with PRR7 (Supplementary Fig. 7c). However, we were unable to examine the effect of PRR7 on the surface expression of GPR177 due to the lack of good antibodies suitable for the immunostaining. Taken together, these data suggest the possibility that PRR7 interacts with GPR177 to prevent its function of Wnt secretion. We described the data in the paper and provided a brief discussion by inserting a line that “our data suggest that PRR7 inhibits Wnt secretion by potentially interfering with GPR177 function”.

4. Some minor comments:

Line 93: What are APV, CNQX? Please specify in the text.

– The full names of these pharmacological agent and their actions are now detailed in the text: “...NMDA receptor antagonist (2-amino-5-phosphonopentanoic acid; APV) but not by AMPA receptor antagonist (6-Cyano-7-nitroquinoxaline-2,3-dione; CNQX) or GABA_A receptor antagonist (bicuculline; Bic)..”.

Line 183: in or on exosomes

– Modified the text to “PDZ-interaction is not required for the secretion of PRR7 on exosomes but...”.

Line 252: Do you mean Figure 6a (inset)?

– Yes and corrected.

Figure 6 Quantification of PRR7 overexpression effect on polyubiquitination in Fig 6c or mention in the text as in Fig 6e.

– The effect of poly-ubiquitination of total neuronal proteins (K49-pUb) is shown in Fig. 7c and quantified in Fig. 7d. On the other hand, Fig. 7e refers to the effect on the poly-ubiquitination of PSD-95.

Line 402 PRR7 – Corrected.

Line 573 Which company provided active Wnt5a and Wnt7a?

– They were obtained from R&D Systems and now specified in the Methods section.

Reviewer #3 (Remarks to the Author):

The paper presented an interesting set of data on PRR7 regulating synapse development. PRR7 was shown to be in exosomes and inhibit synapse formation. Analysis of synapses by morphology and electrophysiology experiments appeared to be solid. PRR7 was shown to act by blocking Wnt secretion from exosomes (thus activating GSK3b and PSD degradation). The hypothesis appeared to be novel and may be useful to peers in the field. There are a few points that need to be tightened up prior to publication.

1. The title “Reciprocal Control of Excitatory Synapse Numbers by Wnt and Novel Wnt Inhibitor Secreted on Exosomes” is superficial and misleading. It does not even mention PRR7.

– We changed the title to “Reciprocal Control of Excitatory Synapse Numbers by Wnt and Novel Wnt Inhibitor **PRR7** Secreted on Exosomes”

2. Wnt release from NMJ is not an accurate statement. The cited papers were dealing with *Drosophila* NMJ, not mammalian NMJ. These two synapses used completely different neurotransmitters.

– We changed the text to indicate specifically *Drosophila* NMJ.

3. Figure 1. Punctas or dots “outside” of neurons – could they be axons or dendrites of unlabeled neurons? Do they colocalize with PSD95 or other synaptic markers? How about exosome markers?

– To address the questions, we performed additional experiments. In addition to dendrites and axons of un-transfected neurons, these puncta frequently localized to “empty” extracellular space (perhaps by non-specific binding to poly-D-lysine/laminin substrates), as shown by co-immunofluorescent staining with Tau (axon marker) and MAP2 (dendritic marker) (Supplementary Fig. 1a) or PSD-95 (Supplementary Fig. 1b). These data are consistent with the fact that PRR7 could be purified from culture supernatant in association with exosomes. We also tried the suggested colocalization experiments with two exosomal markers, flotillin-1 and Alix, but, unfortunately, antibodies recognizing these markers did not produce specific staining in immunocytochemistry (data not shown), although they worked well for western blotting.

4. What is the evidence that the vesicle in EM image is the one containing PRR7? Such claim cannot be made unless immunoEM is done. EM images of vesicles where PRR7 is low should be provided to conclude that PRR7-containing vesicles are unique.

– Immunogold-labeling of exosomal proteins localized to the intravesicular space of exosomes requires the permeabilization of exosomal membranes, which is difficult to control and affects the morphology of exosomes (They et al., 2006). Since we expected that the epitopes of PRR7 antibodies (for both HA-tag and endogenous) faces the intravesicular side of exosomes (Supplementary Fig. 5), we used saponin (0.05%) to permeabilize the exosomal membranes for the immunoEM experiments. We found consistent immunogold-labelings of electron dense-vesicular looking structures of ~ 80-90 nm in diameter (Fig. 1e, top two panels). In the absence of primary antibodies or the saponin-permeabilization, we did not observe the immunogold-labelings (Fig. 1e, the bottom panel and Supplementary Fig. 1b). These data suggest that PRR7 is indeed secreted on exosomes and further confirm the membrane topology of PRR7 in exosomes. However, we did not find any evidence that PRR7-containing exosomes are unique and distinguished from other exosomes in morphological terms.

5. Biochemical fraction experiments need control markers. It would be important to show that some

markers are not clusters in the middle fractions (where PRR7 is). At the moment, all proteins tested were clustered in the middle fractions.

– We would like to point out that the biochemical fractionation experiments (sucrose gradient) were performed using purified exosomes (100,000 ×g pellet). The purpose of the experiments was to confirm that PRR7 is indeed associated with exosomes by determining their equilibrium densities, which is a standard analytical method in exosome field. We tried to find other markers showing different distribution as the reviewer's suggested but we were unsuccessful. We consider the clustering of the known exosome-secreted proteins (aka, exosome markers) in the middle fractions as a strength indicating the purity of the exosome preparations.

6. Figure 2. Overexpressing PRR7 reduces synapses in non-transfected (Nxf) neurons – Is this a toxic effect or effect of overexpressed PRR7 in transfected (Txf) neurons (both pyramidal and interneurons)? This should be addressed, considering others have some that PRR7 is pro-apoptotic. The authors have shown that neurons are not dead, but were they healthy? If not, synapses may disappear.

– Under our experimental conditions, PRR7 did not show pro-apoptotic effect in neurons (Supplementary Fig. 3). In addition, these neurons are morphologically indistinguishable from GFP-transfected neurons in terms of dendritic tapering and arborizations (Supplementary Fig. 1a and 3a). We found that the overexpression of PRR7 reduces excitatory synapses in both glutamatergic (pyramidal) and GABAergic interneurons (data not shown). Importantly, PRR7 overexpression did not affect inhibitory synapses in Nxf neurons (Fig. 4), indicating specific effect. Additionally, the caspase inhibitor z-DEVD-FMK (blocks apoptosis) did not block the effect of PRR7 in Txf and Nxf neurons (Supplementary Fig. 3g). Finally, patch clamping experiments of neurons are very sensitive to neuronal health. We did not find a difference in the success rates of patch clamp recordings between control (GFP) and PRR7-transfected neurons. Therefore, it is highly unlikely that the reduction of synapse numbers in PRR7 transfected and neighboring neurons is a non-specific effect due to the general sickness of neurons.

7. As shown in later figures, PRR7 overexpression in pyramidal neurons and in interneurons alter their activity. Thus, the effect on Nxf neurons could be a secondary effect? Can this effect be blocked by exosome inhibitors?

– To the best of our knowledge, there are no specific inhibitors of exosomes. Instead, we used culture supernatant depleted of exosomes to show that the effect on Nxf neurons are mediated by exosomes containing high levels of PRR7 (please see below our responses to #9 point).

8. In various places, PRR7-enriched exosomes were referred. It would be important to include PRR7-poor exosomes as control.

– We realize that “PRR7-enriched exosomes” could be misleading – we used the term to refer to the exosome preparations purified from PRR7 overexpressed neuron culture (therefore, to be exact, exosome preparations containing higher amount of PRR7 proteins than normal exosomes). We clarified this in the text and changed to “exosomes with high levels of PRR7”. In addition, we used the culture supernatant or exosome preparations from GFP-transfected neurons as controls (Fig. 5a-d, h), representing culture supernatant (containing exosomes) or exosomes secreted under normal growth conditions.

9. Can PRR7's effect be tested directly by tossing purified, recombinant PRR7 or different domains of it on neurons? This is important as it will change the conclusion. PRR7 could be a marker of exosomes and

may be necessary for exosomes' effect on synapses, but not PRR7 itself directly regulates synapses. Along the same line, do exosomes of HEK293 cells have similar effect to those of Txf neurons? Why not? Don't they express Wnts?

– These are interesting ideas and important questions. To address them, we performed additional experiments. First, per this reviewer's request, we tested the effect of purified full-length GST-tagged PRR7 protein. As shown in Supplementary Fig. 4, the application of purified GST-PRR7 (250 ng/ml; > 10× greater than the amount of PRR7 present in neuronal exosomes purified from the culture supernatant [CS] of HA-PRR7 over-expressed neurons) to the hippocampal neuron culture had no effect on PSD-95 puncta. These results excluded the possibility that parts or full-length PRR7 secreted independently of exosomes exerts effect on neighboring neurons. Second, we tested CS of PRR7-transfected neurons that was devoid of exosomes (by centrifugation at 100 k ×g for 3 h). Unlike the complete CS (PRR7 CS), CS devoid of exosomes (CS-Exo) had no effect on PSD-95 clusters (Supplementary Fig. 4a,b), indicating that PRR7-containing exosomes produce the effect.

If PRR7 is simply a marker of exosomes, it is expected that the increase in the amount of PRR7 in exosomes by HA-PRR7 transfection would also increase the amount of conventional exosomal markers Flotillin-1 and Alix in exosomes. However, the amount of these markers in exosomes remained the same (Fig. 5e, 7c and Supplementary Fig. 7b), arguing against the idea.

Interestingly, exosomes purified from the CS of PRR7-overexpressing HEK293 cells did not have a significant effect on PSD-95, despite the fact that they do contain a plenty of PRR7 (Supplementary Fig. 4). However, the results are not too surprising, considering the previous studies showing that exosomes produced in heterologous N2a cells are uptaken by microglia but not by neurons (Yuyama et al., JBC, 2012). On the other hand, neuronal exosomes are only uptaken by neurons and not by glia (Chivet et al., J. Extracellular Vesicles, 2014). These papers indicate that exosome uptake has cell-type specificity. Therefore, the lack of effect by PRR7-containing HEK293 exosomes could be explained by the possibility that neurons do not uptake HEK293 exosomes. Therefore, taken together, these data strongly suggest that PRR7 is the major molecule in exosomes that causes excitatory synapse loss in neurons. Nonetheless, we noted in Discussion that “it remains a possibility that PRR7 might promote co-transport of unidentified specific molecules (RNAs or proteins) in the exosomes to exert its effect on neighboring neurons”.

10. The linear pathways of PRR7-Wnt seemed to be over-simplified. The evidence that Wnts work directly downstream of PRR7 is weak although they could “rescue” the phenotypes. There could be many steps in between.

– We agree with this reviewer's view and thus removed the word “directly” in the text to eliminate the confusion that we are alluding to an idea that Wnts work directly downstream of PRR7. We also suggested that Wnts and PRR7 work in “mutually opposing” function for the regulation of synapses in the manuscript.

11. Wnt5a was shown to promote the formation of inhibitory synapses (Cuitino, 2010). PRR7's effect was specific for excitatory synapses, but not GABA synapses. However, Wnt5a was believed to be an effector of PRR7. Something does not connect here.

– The findings of Cuitino et al. and ours are not in conflict, as we discussed in detail above (please see our responses to the Reviewer 1's Specific Comments #8).

Reviewers' comments:

Reviewer #1 (Remarks to the Author):

In the revised manuscript and rebuttal letter, the authors addressed many of the concerns raised by this reviewer. However, I am very disappointed by the lack of thoroughness and careful revision of the paper.

Many of the key questions were not properly addressed.

Some of the images are still problematic because they do not clearly show what the authors claim. For example, it is unclear why the authors did not compare properly control cells expressing GFP with cells expressing GFP plus the PRR7, this should be proper comparison. In other experiments, control cells were expressing β -gal and these cells were compared to GFP.

Supplementary figure 2 a: the authors now provided low magnification images to show the level of PSD95 puncta between GFP control and HA-PRR7 cells. Although the field of cells containing HA-PRR7 have less staining for PSD95, the number of cells is significantly lower than the control, which could explain the result in the number of puncta. Better images should be provided.

Point 8, regarding the impact of PRR7 on Wnt5a. The authors wrote "Wnt5a reduction is not expected to affect GABAergic synapses significantly as demonstrated by Cuitino et al (2010) and Marzo et al (2016)". This statement is incorrect. Marzo et al did not examine Wnt5a but Dkk1.

More importantly the original question raised by this reviewer about "how over expression of PRR7 has a moderate effect on inhibitory synapse number on cells that overexpressed this protein but not on neighbouring cells. If the authors are correct that PRR7 is released, then it should affect inhibitory cells. Their explanation in the rebuttal did not make sense to me.

The response to the question about the model where the authors flip the PRR7 protein in the membrane is not satisfactory. The authors agree that changing the membrane topology would be an unlikely event. However, the authors did not change the model at all. This is unacceptable.

Reviewer #2 (Remarks to the Author):

The revised version of the manuscript from Lee and colleagues is substantially improved regarding structure and description of methodology. It is clearly of high interest to readers of Nature Communications.

The experiments connecting PRR7 with Wnt secretion are convincing and definitely interesting. But there is still a critical open point. I agree with reviewer #1 that the topology issue of

PRR7 is confusing and needs very careful statements or better experiments.

Whether in fusion or secretion of exosomes, the membrane topology always remains preserved (luminal=extracellular). MVBs generate ILVs by inward (=negative) budding, so ILVs enclose cytoplasm and have an extracellular outside, as seen in the scheme #R1 -15) in the rebuttal letter. When the MVB fuses with the plasma membrane that is preserved. Similarly, when exosomes reach a target cell, either they would fuse with the plasma membrane or get endocytosed then backfuse with the endosomal membrane, preserving the membrane topology.

A switch in membrane topology of a tagged construct rather indicates that the tag is interfering with the cotranslational integration into the ER membrane. In addition, the membrane topology prediction using TMHMM of PRR7 reveals the C-Terminus to be intracellular, so a type I membrane protein as suggested by previous publication (Hrdinka M et al., JBC 2011). To claim that PRR7 is a type II protein that switches topology during its trafficking onto exosomes is indeed premature and should be omitted from the manuscript.

Reviewer #3 (Remarks to the Author):

The authors have performed additional experiments to address major concerns I had. They have provided explanations to others. I believe that this paper addresses an important question and now support its publication at Nature Comm.

Responses to Reviewers' Comments

First of all, we sincerely appreciate the reviewers for the encouraging and constructive comments on our manuscript. We hope that the revised manuscript alleviates the reviewers' remaining concerns. Modified or newly added texts in the revised manuscript are marked by red font. Our point-by-point responses to the specific comments are below.

Reviewer #1 (Remarks to the Author):

In the revised manuscript and rebuttal letter, the authors addressed many of the concerns raised by this reviewer. However, I am very disappointed by the lack of thoroughness and careful revision of the paper. – We would like to express our regret that we did not provide a sufficiently thorough and careful revision of the paper for this reviewer. We have further revised the manuscript and provided additional explanations and data to address the reviewer's concerns. We hope that this will satisfy any remaining concerns.

Many of the key questions were not properly addressed.

Some of the images are still problematic because they do not clearly show what the authors claim. For example, it is unclear why the authors did not compare properly control cells expressing GFP with cells expressing GFP plus the PRR7, this should be proper comparison. In other experiments, control cells were expressing β -gal and these cells were compared to GFP.

– We used either GFP or β -Gal as a transfection control or as cell-fill maker. β -Gal, like GFP, have been frequently used as controls in similar experiments by us and others (Lee et al., *Neuron*, 2002; Shin et al., *Nat. Neurosci.* 15, 2012; Campbell and Sheng, *J. Neurosci.* 38, 2018). Under the conditions we used, neurons transfected with GFP and β -Gal produced similar results (PSD-95 puncta densities), comparable to non-transfected neurons (data not shown). We also would like to note that we did not directly compare β -Gal (control cells) to GFP in any of our experiments. Of course, we do understand the reviewer's point and we would have used GFP vs GFP + HA-PRR7 in the experiments, if we were simply comparing the differential effects between GFP vs HA-PRR7. The sole reason why we did not use GFP vs GFP + HA-PRR7 vs GFP + PRR7-GFP scheme in Fig. 1a, (in actual experiments, we used GFP + β -Gal, HA-PRR7 + β -Gal vs PRR7-GFP + β -Gal as we previously explained in the Responses to the Reviewer's comments; GFP as a transfection control and β -Gal as a cell-fill marker for tracing dendrites of transfected neurons) is to make sure that GFP-fluorescent cells are expressing PRR7-GFP and not just GFP alone. Therefore, we do not see a problem in the experimental design since GFP expression under the conditions did not produce significant effect on its own. For this reviewer, we now provide a separate data set comparing the results from GFP control cells to GFP + HA-PRR7 transfected neurons. As shown in Figure for Reviewer 1, the data are essentially identical to the data shown in Figure 1a in the manuscript. Therefore, we believe that the data presented in the manuscript do support our main conclusions.

Figure for Reviewer 1. Effect of PRR7 overexpression on PSD-95 puncta density. $n = 20$ from 2 replicates. **** $P < 0.001$.

Supplementary figure 2 a: the authors now provided low magnification images to show the level of PSD95 puncta between GFP control and HA-PRR7 cells. Although the field of cells containing HA-PRR7 have less staining for PSD95, the number of cells is significantly lower than the control, which could explain the result in the number of puncta. Better images should be provided.

– Upon the comment, we counted the number of cells in the two images and found that there was indeed about 10% less neurons for the HA-PRR7 image. We understand the reviewer’s concern and apologize for not paying attention to the fine details. Now we offer a new set of images, which have similar number of cells but shows a clear reduction in PSD-95 staining intensities in HA-PRR7 transfected image.

Point 8, regarding the impact of PRR7 on Wnt5a. The authors wrote “Wnt5a reduction is not expected to affect GABAergic synapses significantly as demonstrated by Cuitino et al (2010) and Marzo et al (2016)”. This statement is incorrect. Marzo et al did not examine Wnt5a but Dkk1.

– This reviewer correctly pointed out that Marzo et al. did not examine Wnt5a directly. However, we referred to the paper to bring out the observation that the blockade of Wnt signaling using Dkk1 overexpression did not affect the number of GABAergic synapses in vivo significantly. Since Dkk1 functions as Wnt inhibitor including Wnt5a, the paper supports our opinion that it is unclear whether Wnt5a is involved in the regulation of GABAergic (inhibitory) synapse numbers.

More importantly the original question raised by this reviewer about “how over expression of PRR7 has a moderate effect on inhibitory synapse number on cells that overexpressed this protein but not on neighbouring cells. If the authors are correct that PRR7 is released, then it should affect inhibitory cells. Their explanation in the rebuttal did not make sense to me.

– We want to clarify that we did not state in the manuscript that PRR7 does not affect neighboring “inhibitory cells”. As we explained above and previously, first of all, there is inconsistency in the literature whether Wnt5a is involved in the regulation of GABAergic (inhibitory) synapse numbers. Perhaps, the reason why PRR7 does not affect GABAergic synapses in neighboring neurons is due to the differences in the molecular mechanisms by which PRR7 exerts its effects on excitatory and GABAergic synapses. Hammering out the details of the differences will take extensive further studies (and thus out of scope of this manuscript). Nonetheless, to suggest one hypothetical possibility, exosomal PRR7 absorbed by neurons might not enter proper subcellular-compartment that is necessary for initiating cellular signaling cascades that lead to the reduction in GABAergic synapses. For these reasons, we stated in the Discussion that “Therefore, it remains to be further investigated whether PRR7 and Wnts control GABAergic synapses under physiological conditions.”

The response to the question about the model where the authors flip the PRR7 protein in the membrane is not satisfactory. The authors agree that changing the membrane topology would be an unlikely event. However, the authors did not change the model at all. This is unacceptable.

– We agree with the reviewers’ opinion that much more studies are necessary to unequivocally determine the membrane topology of PRR7 and that the change in the membrane topology of PRR7 in the proposed model is unusual and unproven at this time. Therefore, we revised the main text and relevant figures (Supplemental Fig. 5) to remove the portions regarding the membrane topology (type II) of PRR7. We also removed the membrane topology indicators (N- and C-term) of PRR7 in the model accordingly.

Reviewer #2 (Remarks to the Author):

The revised version of the manuscript from Lee and colleagues is substantially improved regarding structure and description of methodology. It is clearly of high interest to readers of Nature Communications.

The experiments connecting PRR7 with Wnt secretion are convincing and definitely interesting. But there is still a critical open point. I agree with reviewer #1 that the topology issue of PRR7 is confusing and needs very careful statements or better experiments. Whether in fusion or secretion of exosomes, the membrane topology always remains preserved (luminal=extracellular). MVBs generate ILVs by inward (=negative) budding, so ILVs enclose cytoplasm and have an extracellular outside, as seen in the scheme #R1 -15) in the rebuttal letter. When the MVB fuses with the plasma membrane that is preserved. Similarly, when exosomes reach a target cell, either they would fuse with the plasma membrane or get endocytosed then backfuse with the endosomal membrane, preserving the membrane topology. A switch in membrane topology of a tagged construct rather indicates that the tag is interfering with the cotranslational integration into the ER membrane. In addition, the membrane topology prediction using TMHMM of PRR7 reveals the C-Terminus to be intracellular, so a type I membrane protein as suggested by previous publication (Hrdinka M et al., JBC 2011). To claim that PRR7 is a type II protein that switches topology during its trafficking onto exosomes is indeed premature and should be omitted from the manuscript.

– We agree with the reviewers' opinion and removed the relevant text and topology markers in the model (please see our responses to the last comments from the reviewer 1 for details.)

Reviewer #3

(Remarks to the Author): The authors have performed additional experiments to address major concerns I had. They have provided explanations to others. I believe that this paper addresses an important question and now supports its publication at Nature Comm.

– We appreciate the reviewer's positive evaluation on the revised manuscript that we addressed his/her major concerns and provided explanations to others.

REVIEWERS' COMMENTS:

Reviewer #1 (Remarks to the Author):

The authors has addressed all the concerns raised by this reviewer.

I would recommend publication in Nature Communications.

Reviewer #2 (Remarks to the Author):

The authors have addressed my concerns and I can now fully support publication in Nature Communications of this manuscript.